# Cytoplasmic sequestering of a fungal stress-activated MAPK in response to a host plant phenolic acid

**Rina Zuchman[1,2,3,4], Roni Koren[1], Tamar Ziv[2], Yael Lupu-Haber[5], Nitsan Dahan[5], Ofri Levi[1,3,4]\*, Benjamin A. Horwitz** [1]\*

1 Faculty of Biology, Technion - Israel Institute of Technology, Haifa, Israel, 2 Smoler Protein Center, Technion - Israel Institute of Technology, Haifa, Israel, 3 MIGAL - Galilee Research Institute, Kiryat Shmona, Israel, 4 Faculty of Sciences and Technology, Tel-Hai Academic College, Tel-Hai, Upper Galilee, Israel, 5 Life Sciences and Engineering Infrastructure Center, Lorry I. Lokey Interdisciplinary Center, Technion - Israel Institute of Technology, Haifa, Israel

\* horwitz@technion.ac.il (BAH); ofrilevi88@gmail.com (OL)

OPEN ACCESS

## Abstract

Fungal pathogens employ conserved signaling pathways to survive in the host. The stress-activated MAP kinase Hog1 of the maize pathogen *Cochliobolus heterostrophus* undergoes dephosphorylation upon exposure to ferulic acid, a phenolic compound abundant in the host plant. Unlike its nuclear localization during osmotic stress, Hog1 forms cytoplasmic foci in response to FA, indicating its sequestering to a compartment or condensate. FA prevents several characteristic responses of the Hog1 pathway to osmotic stress: hyperphosphorylation of Hog1, nuclear localization, and expression of a monosaccharide transporter gene, *MST1*. Under FA stress, mRNA-containing foci are formed, as visualized by sm-FISH. Hog1 foci extensively colocalize with mRNA foci. Hog1 did not colocalize with nuclei or peroxisomes. Fragmented mitochondria, appearing upon FA exposure with a delayed time course compared to Hog1 sequestration, were mostly distinct from the Hog1 foci, with few instances of colocalization. With tagged Hog1 as an affinity purification bait, we isolated an FA-dependent sub-proteome from a subcellular fraction enriched with fluorescent foci. The identified proteins include RNA-binding proteins, translation initiation factors and mitochondrial proteins. The RRM and pumilio domain protein Puf2 was enriched, and live imaging confirmed the accumulation of a Puf2 fluorescent fusion and its colocalization with Hog1 foci following FA induction. Stress-induced sequestering of MAPK Hog1 to RNA-containing granules, together with dephosphorylation, has the potential to collectively promote survival on the plant host where stress might cause over-activation of Hog1. Conversely, FA as a host defense interferes with stress-induced Hog1 nuclear localization and downstream gene expression. The MAPK signaling mode defined by the response of Hog1 to FA is thus relevant to both host defense and pathogen survival.

**Data availability statement:** Data availability: all data are included in the manuscript and Supplementary Material. The mass spectrometry proteomics data have been deposited to the ProteomeXchange Consortium via the PRIDE partner repository with the dataset identifier PXD044457. Fungal strains are available from the Faculty of Biology, Technion (contact: biheadadmin@technion.ac.il).

**Funding:** This study was funded by Israel Science Foundation (ISF) grants 2381/15, 420/17 and 927/22 to B.A.H. R.Z. predoctoral and O.L. postdoctoral fellowships and R.K. salary were supported in part by ISF 927/22, funder URL: https://www.isf.org.il. Microscopy at the Life Sciences & Engineering Center was funded in part by the Russell Berrie Nanotechnology Institute, Technion, in the form of discounted equipment use, to B.A.H. and members of the Horwitz lab, funder URL: https://rbni.technion.ac.il. The funders did not play any role in the study design, data collection and analysis, decision to publish, or preparation of the manuscript.

**Competing interests:** The authors have declared that no competing interests exist.

## Author summary

Fungal pathogens respond to signals from their host and environment. Processing this information lets them reprogram gene expression and metabolism to survive stresses and host defenses. A host antimicrobial compound, the simple phenolic ferulic acid, kills cells of the plant pathogen responsible for southern corn leaf blight. A stress response protein conserved among eukaryotes, the MAP kinase Hog1, usually transmits signals when it is phosphorylated, and does so by accumulating in the nucleus to turn on genes that promote survival and growth under stress. Here we found a novel signaling mode, where Hog1 is sequestered away from the nucleus in cytoplasmic granules. This prevents the normal stress response, so that ferulic acid could help the host stop progression of disease. On the other hand, over-activation of Hog1 promotes cell death, so sequestering could be a survival response of the pathogen. Either or both mechanisms could be at work in this and other pathosystems.

## Introduction

Pathogens detect host-derived molecules as signals, providing information on the presence of the host, or of impending stress generated by host defense. Most filamentous fungal genomes encode three MAP kinases. Of these, Hog1, the high osmolarity glycerol pathway regulator first discovered in yeast, is central to osmoregulation as its name implies, but is also involved in sensing multiple stresses. The closest mammalian ortholog is P38. Hog1 acts downstream of a branched two-component system sensory pathway for osmoregulation, converging on dual phosphorylation at the TGY motif that is conserved in Hog1/P38 orthologs (reviewed by [1]. More recent work continues to uncover variations on the canonical pathway, to give one example, the finding that in *Candida albicans* the rate of dephosphorylation is more important than phosphorylation by upstream kinases for the response to oxidative stress [2]. Study of the role of Hog1 in the response of the fungal pathogen *Cochliobolus heterostrophus*, agent of southern corn leaf blight to ferulic acid (FA), an abundant phenolic in the host plant, maize, has likewise uncovered some novel features. FA triggers a rapid form of regulated cell death (RCD) [3]. Phenolic acids are among the barriers that plants deploy against pathogens. Fungal pathogens have evolved the ability to degrade diverse plant defense compounds, which drive selection of metabolic pathways in fungal genomes [4,5]. In Ceratocystidaceae genomes for example, catechol dioxygenases are correlated with pathogenicity [6]. Phenolics are nevertheless carbon sources for fungal growth, particularly those with vicinal OH groups on the phenolic ring. Phenolics lacking this catechol structure are more toxic, less readily degraded, and perceived by the pathogen as a stress [7]. Loss of the stress-activated MAPK ortholog ChHog1 (High Osmolarity Glycerol 1) of *C. heterostrophus* reduces virulence and affects viability in liquid culture, while it is dispensable for mating and conidiation [8]. ChHog1 is required for full tolerance to FA [3]. Perhaps

surprisingly, ChHog1 was found to be dephosphorylated in response to FA [3], in contrast to activation by dual phosphorylation under osmotic stress. We proposed [3] that lowering the (relatively high) basal level of Hog1 phosphorylation might mitigate RCD, noting that in yeast sustained activation causes growth arrest or cell death [9,10]. Indeed, the fungicide fludioxonil kills fungal cells by Hog1 hyperactivation [9,11,12]. A stress-sensitive transcription factor (ChAP1) also participates in the FA response. Comparison of the transcriptomic signatures of wild-type and *chap1* mutants suggested that when stress overcomes the ability of the cell to respond, it initiates regulated cell death (RCD) [13]. Cells have multiple ways to avoid RCD and return to homeostasis [14]. Dephosphorylation, attenuating Hog1 signaling, could mitigate FA-promoted cell death. We reasoned that preventing the active dually phosphorylated MAPK from reaching its targets could contribute as well and therefore followed the intracellular localization of a C-terminal Hog1:GFP fusion, discovering ChHog1 accumulation in cytoplasmic foci in response to a host defense compound. Hog1 is needed, however, for wild-type tolerance of FA [3], seemingly at odds with the idea that over-activation is a driver of cell death. Furthermore, the host could have evolved to expose the pathogen to defense compounds that suppress the ability to respond to stress. The aims of this study were thus to explore both these modes of FA action on the fungal pathogen, and to predict how they could affect its survival and its tolerance of host defenses.

## Results

### Hog1 localization in response to stress

To follow the cellular localization of *C. heterostrophus* Hog1 (ChHog1) in response to FA stress, we replaced the resident copy with a C-terminal Hog1:GFP fusion by a double-crossover event, retaining the native upstream regulatory region at the same genomic location. The gene replacement did not affect growth, development or virulence (S1A and S1B Fig). The fusion protein appeared in immunoblots at the expected size. To assess functionality of the fusion protein, we chose conditions for osmotic stress and exposure to FA from previous studies on *C. heterostrophus* [3,15]. Dual phosphorylation of the fusion protein in response to osmotic stress by 1 M sorbitol as well as dephosphorylation in response to 2.5 mM FA were similar to native WT Hog1 in high osmolarity conditions: Hog1 was strongly phosphorylated within 10 min, and dephosphorylated within 10–20 min from the start of exposure to FA as previously reported [15] (S1C Fig). Hog1 translocated to the nucleus upon exposure to high osmolarity as expected, indicating a functional Hog1:GFP fusion protein (Fig 1A and 1B). Surprisingly, upon exposure to 2.5 mM FA, Hog1:GFP did not translocate to the nucleus, but rather accumulated in cytoplasmic foci (Fig 1A and 1B). The distribution of constitutively expressed GFP alone (under *Aspergillus nidulans* Pgpd) did not change in response to osmotic stress or exposure to FA (S1D Fig). Lower levels of FA often induced full conversion of Hog1:GFP in a given hyphal compartment to the focal distribution, leaving neighboring compartments unaffected, implying a concerted transition (S1E Fig). These experiments defined the concentrations and time frame to follow intracellular localization of Hog1:GFP.

To test the time dependence of granule formation, and the possibility that ChHog1 could initially be translocated to the nucleus upon exposure to FA and then aggregate in cytoplasmic foci, Hog1 localization was observed at different times. FA-induced recruitment to foci was also compared to high osmolarity and heat shock stress responses. Upon osmotic stress, Hog1:GFP strongly localized to the nuclei within 10 minutes. Upon either FA or heat shock treatments ChHog1 was detected in the nucleus at 10 minutes from the start of stress, at a level consistent with that seen in many samples prior to stress (Figs 1A and S2, top row). During heat treatment, ChHog1 started accumulating in cytoplasmic foci within 30 minutes, and by 60 minutes was observed only within foci. One hour after relieving the heat shock, ChHog1 dispersed back to the cytoplasm (S2 Fig, bottom row). Exposure to FA caused fasterHog1:GFP localization to bright cytoplasmic foci, starting within 10 minutes. The pattern observed is very similar to that seen at later times after heat shock (30 and 60 min, S2 Fig). Caffeic acid differs from ferulic acid only in that there is an OH at position 3 instead of the methoxy group present in FA, is metabolized via the β-ketoadipate pathway but otherwise has similar properties, serving as a negative control. Caffeic acid, which does not induce RCD-like cell death [3] did not trigger Hog1:GFP localization to foci (S2 Fig).

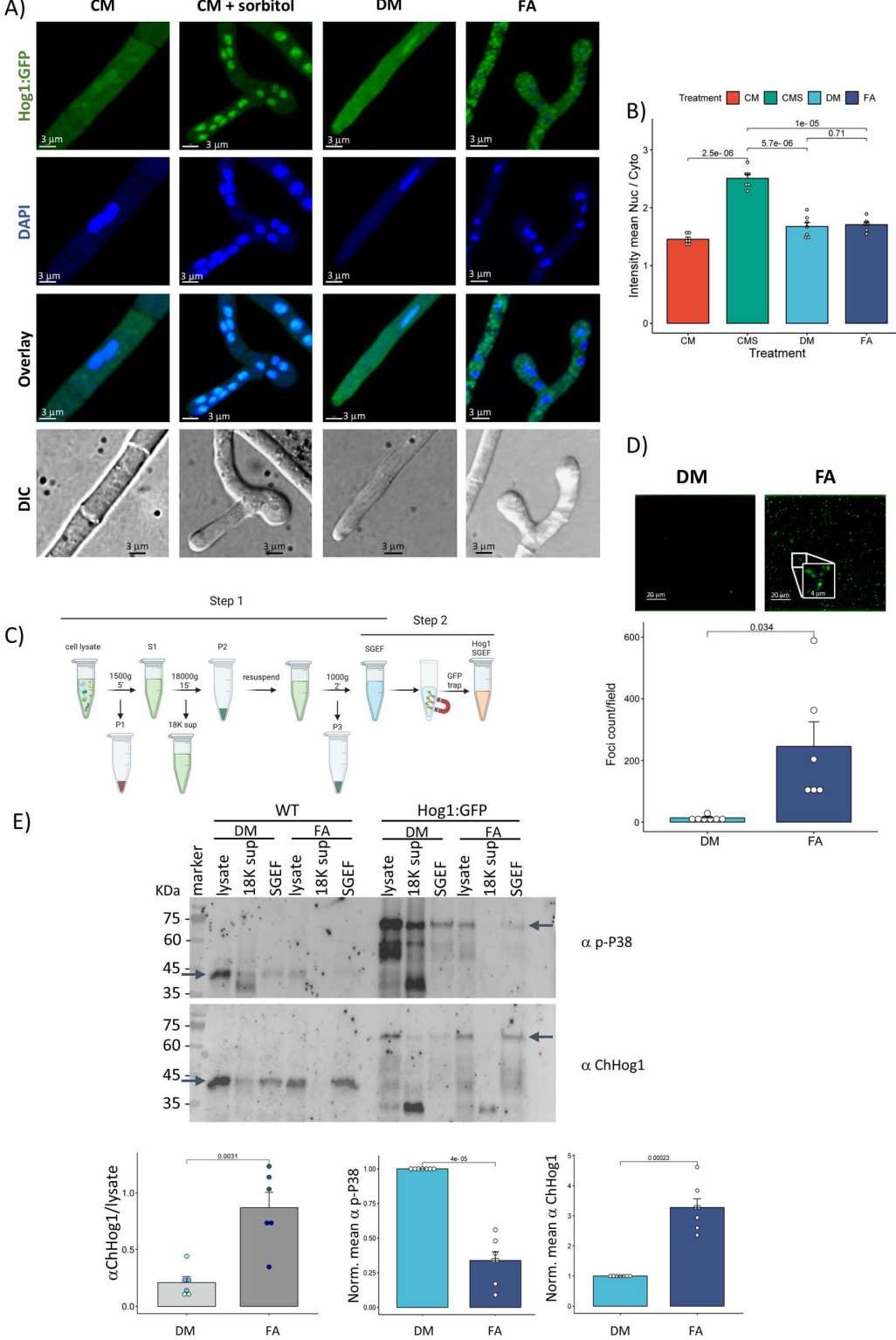

**Fig 1. Localization of Hog1:GFP fusion protein. A)** Cells expressing Hog1:GFP fusion protein were grown on selective plates for 2 weeks. $10^5$ spores collected in water were set in a 60 mm plate containing cover slips, incubated 5-6 h for germination and adherence to the glass surface, and the water replaced with complete medium (CM). After a further 16 h incubation CM was replaced with fresh CM (control), CMS (CM supplemented with 1

M sorbitol) for 10 min at RT for osmotic stress, left panels. For FA exposure, right panels, CM was replaced with CM supplemented with DMSO, solvent for FA (DMSO) or CM supplemented with 2 mM ferulic acid (FA) in DMSO for 30 min, RT. Cells were then fixed with 4% paraformaldehyde in PBST and imaged with a spinning disk confocal microscope (Yokogawa CSU-W1, Nikon). Scale bars, 3 µm, for convenience a 15 µm scale bar is shown below the composite. Images are representative of multiple repeats (n > 10). Green, Hog1:Gfp; Blue, DAPI. Confocal images are z-stacks; bright field images are not confocal. Different numbers of nuclei in the hyphal compartments shown are not related to the treatments and depend on the hyphal segment viewed (compare S2C Fig). **B)** Quantitation of nuclear retention of Hog1:GFP in treatments as shown in **(A)** (calculated from 3 repeats, 6 fields). The P values indicated are for pairwise t tests. **C)** Scheme of Hog1:GFP crude foci (SGEF) isolation protocol and Co-IP. Cells were grown and treated with FA as in **(A)**, collected from the plate surface with a cell scraper and extracted with native buffer (see Methods). Foci were enriched by stepwise centrifugation (step 1). Co-IP was performed starting with the SGEF (step 2). Created in BioRender. Horwitz, **B.** (2025) https://BioRender.com/qoqmxas. **D)** Representative confocal images and counts measurements of SG in fraction (multiple repeats; n > 10) from control and FA-treated cells. Scale bar: 20 µm (insert: edge of the frame indicates 4 µm). Bar graph at the right shows the number of granules per microscope field, average and SEM of 6 experiments. The t test probability is indicated. **E)** Top panel: immunoblot probed with a p-P38 or a ChHog1 anti ChHog1, showing relative enrichment for dual-phosphorylated and total ChHog1. The fractions on the blot are labelled according to the isolation protocol in panel **C**. The lanes are, from left to right for WT and Hog1:Gfp strains: lysate (cell lysate, panel **C**), 18K sup (18,000xg supernatant, 18K sup, panel C) and SGEF (SG-enriched fraction, which is the first tube in step 2, labelled SGEF in panel **C**). Sample sets from DMSO control or FA-treated mycelia are labelled DM and FA, respectively; the blot shown is representative of 3 biological repeats from each strain. Loading of the lanes was by equal volume. Lower left, quantitation of ChHog1 signal is shown with the average and SEM for 6 experiments, three for each strain: data for Hog1:GFP (blue symbols) and C4 (green symbols) showed a similar pattern and were pooled; the t test (2-tailed) probability is indicated. Lower center and right panels, quantitation of ChHog1 signal in the SGEF, normalized to the DMSO controls, for a p-P38 and a ChHog1, respectively; average and SEM of 7 biological repeats; t test (two-tailed) probabilities are indicated for each pairwise comparison.

Reaction to FA was robust, ChHog1 remaining in the foci throughout the 60 min stress, and after relieving the stress by replacing the FA-containing medium (S2 Fig, bottom row). Exposure to FA after sorbitol-induced nuclear localization again resulted in formation of fluorescent foci, while exposure to sorbitol after ferulic acid did not relocalize Hog1:GFP from the foci to nuclei (Figs 5, S3A and S3B). The response to FA is thus functionally dominant over osmotic stress. Release of Hog1:GFP from the nuclei when osmotic stress is relieved is rapid (5–10 min, S3A Fig). Sequential osmotic and FA stress resulted in a faster response to FA, with foci visible already after 5 min exposure to FA (S3A Fig), while upon exposure to FA alone, foci begin to appear at 10 min and are fully apparent by 20–30 min.

### Characterization of Hog1:GFP foci

The similarity of the foci to those seen during heat shock (S2 Fig) suggested that ChHog1 is localized to liquid-liquid phase separation (LLPS) condensates upon exposure to FA. To investigate the composition of the foci, we isolated a subcellular fraction by a differential centrifugation protocol designed to enrich stress granules (SG) as described by Jain et al. [16]. We refer to this fraction as SGEF (SG Enriched Fraction) since it contains any other structures or complexes sedimenting under the same conditions (Fig 1C and 1D). The same fractionation was done with WT cells, and the distribution of ChHog1 between fractions, assayed by immunoblot, was similar to that found with the Hog1:GFP strain. The subcellular redistribution is therefore independent of GFP tagging (Fig 1E).

The SGEF appears, visually, enriched for Hog1:GFP fluorescence, containing fluorescent puncta that are of similar size (about 0.5 µm) to the cytoplasmic foci seen in the cell. There are more such fluorescent particles in fractions isolated after FA than in controls (Fig 1D). SGEF is significantly (FA *vs* control) enriched in both Hog1:GFP and the SG marker Puf2, compared to cytoplasmic, ER, nuclear, peroxisomal and mitochondrial conventional markers (S9 Fig). After FA treatment, relative to the control, ChHog1 is more abundant in the SGEF and dephosphorylated (Fig 1E). Control samples are not totally devoid of (phosphorylated) ChHog1 signal in the SGEF; the signal could be from pelleted nuclei (and thus phosphorylated), or other pelletable particles that have affinity for ChHog1. Conversely, after FA treatment, phosphorylated ChHog1 appears at a low level relative to the control in SGEF, due to dephosphorylation, while probing with antibody to total ChHog1 gives a stronger signal (Fig 1E). Proteomic analysis of this fraction (Figs 2A, S5A–S5C) identified 693 proteins that were significantly elevated in the SGEF (out of 2510 identified by at least two peptides) compared to the control (S2 Table, Fig 2A). The FA-elevated proteins include, as expected, many functional classes. These include known

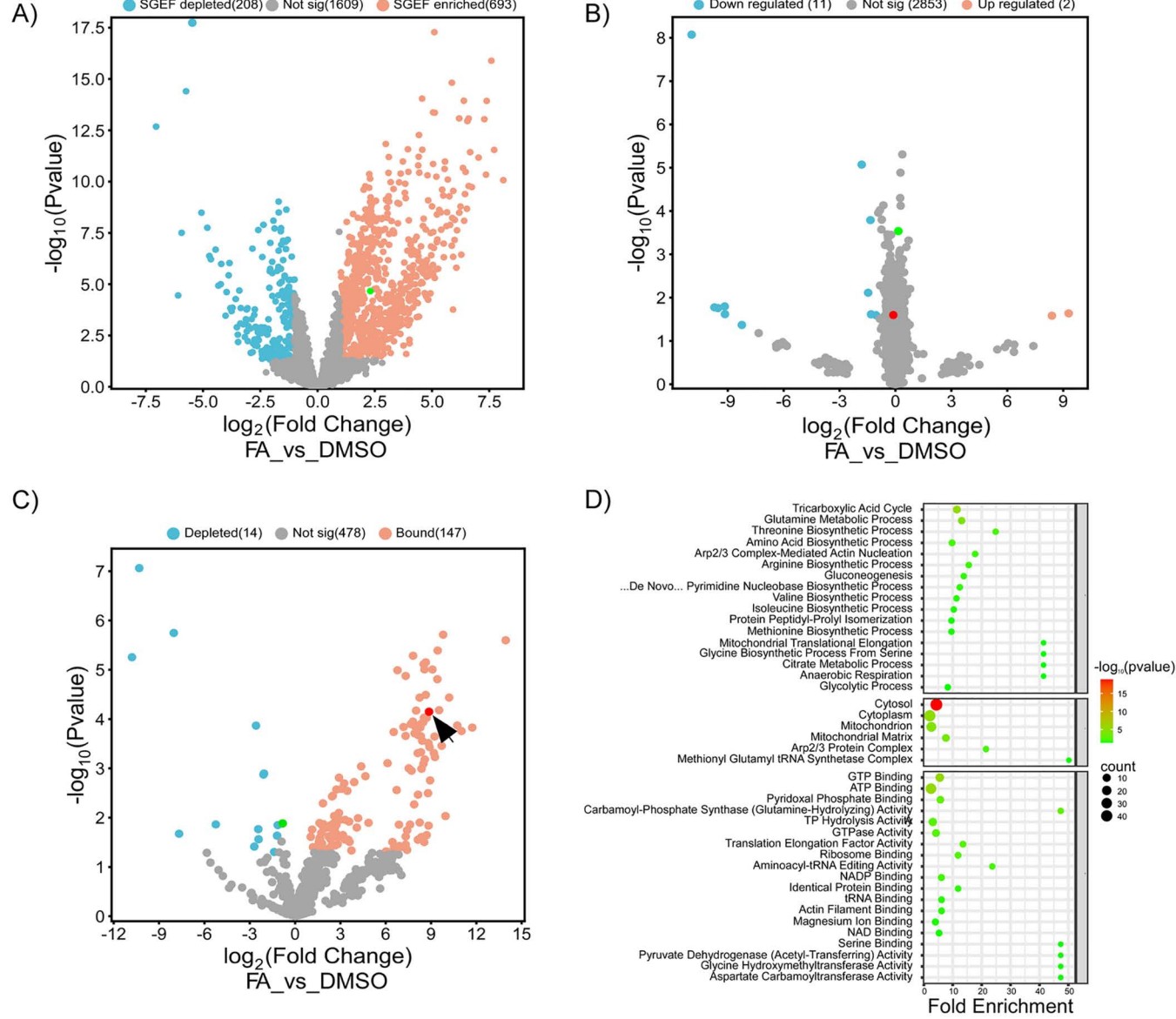

**Fig 2. Proteomic profiling and Hog1 interactome in response to FA treatment.** *C. heterostrophus* cells expressing Hog1:GFP fusion protein were treated with either DMSO (CM supplemented with DMSO, solvent control) or FA (CM supplemented with DMSO and 2.5 mM ferulic acid) for 30 min at RT. Cells were collected, and extracted with native buffer (see Methods) and proteins were analyzed using MS spectrometry. In the volcano plots, each dot represents a protein, with the x-axis showing log2 fold-change (FA *vs.* DMSO) and the y-axis −log10(P-value). Proteins enriched in FA (P < 0.05, log2 fold-change > 1) are marked in orange, DMSO-enriched proteins are marked in turquoise. Hog1 is marked in green and Puf2 (UniProt ID: N4WW71) in red. **A)** SGEF-enriched proteomics. **B)** Total proteomics. **C)** Hog1 Co-IP. **D)** GO term analysis of Hog1 FA interactome. Enriched GO terms (Molecular Function, Cellular Component, Biological Process) are shown with fold enrichment (x-axis) and protein counts (circle size). Analysis performed using DAVID (v2023q2). **E)** Multi-panel scatter plots visualizing proteomic differences across experimental conditions. Each panel represents pairwise comparisons of proteomic differences between: Total proteome Co-IP (log2FC FA-DM), SGEF Co-IP (log2FC FA-DM), SGEF Co-IP (log2FC Hog1:GFP-Untagged C4). Points are colored according to the SGEF log2 FC values using an orange-to-gray-to-purple gradient, where orange represents the lowest values and purple the highest. Puf2 (UniProt ID: N4WW71) and eIF2 (UniProt ID: N4WKP0), are marked.

stress-granule-associated proteins, for example: RNA processing, P bodies and translation-related proteins and MCM complex proteins. RNA binding proteins include known stress-granule markers such as MCM proteins, Tif1, Ded1(15), eIF2 and Jsn1/Puf2 [17] (S2 Table). To test FA impact on protein levels, proteomics of total cell extract was performed. In contrast to the SGEF, only two proteins were FA-elevated in the total proteome (Fig 2B). Specific detection of FA-enriched proteins in the SGEF (Fig 2A) strikingly reflects the sequestering of proteins whose localization is FA-dependent. This sub-proteome is not evident in Fig 2B because the abundance of these proteins is relatively constant over the 30 min FA treatment, only their localization is affected.

To characterize the FA dependent ChHog1 interactome, we performed Co-IP on extracts from the GFP-tagged strain with anti-GFP magnetic beads. First, Co-IP of the total proteome identified 147 FA-enriched proteins (Fig 2C). GO term analysis of these highlighted many functional classes, notably translation-related classes (Fig 2D). The SGEF collected from FA and DMSO-treated cells was also analyzed by Co-IP (Fig 3). 357 proteins were FA-enriched in this fraction (Fig 3A). A SGEF extracted from the untagged WT strain exposed to FA served as an additional control to take into account non-specific interactions with the beads (Fig 3B). 290 proteins were FA-enriched in the Bound samples from Hog1:GFP compared to the Untagged (C4) strain. Ninety proteins passed a double-sieve selection of being enriched compared to both controls (Figs 3B, 3C and S5B). For 47 of these 90, Co-IP gave stronger enrichment (FA vs control) than observed in the initial crude SGEF (S5C Fig). Molecular Function Gene Ontology (GO) term enrichment analysis (using DAVID bioinformatics analysis online tool (v2023q2)) among the 90-protein set revealed enrichment of RNA binding proteins, particularly RNA recognition motif (RRM) - containing; ATPase activity, 5-methyltetrahydropteroyltriglutamate-homocysteine S-methyltransferase activity, translation initiation factor activity and ATP binding (Fig 3D, S2 Table). We used SGD BLASTP to find the yeast homologs of the enriched proteins. Overall, 77 proteins have significant yeast homologs. Twelve of those proteins (~15.5%) were previously reported to assemble into SG upon heat shock (HS) [16,18]. This is a significant enrichment (4.12 fold, hypergeometric p-value = 2.6e-5) compared to a random sample. Three scatter plots which in order compare the enrichment upon FA treatment relative to control across all Co-IP datasets, are shown in Fig 3E. The orange to purple gradient indicates, for each protein, $\log_2$ fold change in abundance for FA relative to the control in the SGEF. The location on each plot compares the relative efficiency of Co-IP in each case. Notably Co-IP from the total proteome and from the SGEF are correlated (left panel). Comparing the normalized untagged WT strain Co-IP dataset (y-axis) with either the total proteome or the SGEF, there is no strong correlation. Nevertheless, enrichment upon FA treatment of SG markers is clear. The correlation between the known SG component Puf2 and Hog1:GFP is evident in all. eIF2 behaves similarly; although less striking, its enrichment is significant (Fig 3E). The enrichment of Hog1:GFP together with SG components provides proteomic evidence that ChHog1 is indeed sequestered to SG.

## Colocalization of Hog1:Gfp with SG components

Co-enrichment of ChHog1 with SG markers supports the hypothesis that ChHog1 is sequestered in or docked to SG or related subcellular condensates. SG typically contain untranslated mRNA as a major component. sm-FISH indicates rapid sequestering of polyA+ RNA to cytoplasmic foci in the wild type, Δhog1 mutant, and when ChHog1 is expressed as a GFP fusion (Figs 4A and S6A). This is characteristic of stress granule formation and consistent with FA acting as a stress. Sequestering of RNA to granules is thus independent of ChHog1, occurring normally in the Δhog1 knockout (S4B Fig). The sm-FISH and Hog1:GFP signals extensively overlap (Fig 4A). The fraction of coinciding or overlapping spots averaged 0.36 over three image sets (Fig 4B). The patterns appear identical in some cells as in the example shown in Fig 4A, while in others there is only partial overlap. A simple interpretation is that mRNA and Hog1:GFP are recruited independently, perhaps by different mechanisms and/or with distinct time dependence. The distribution of coinciding spots over the sampled images (Fig 4B) are consistent with this. The frequency of overlapping spots implies that there is a population of granules labelled by Hog1:GFP that does not contain detectable mRNA, and conversely, a population that contains mRNA but no detectable Hog1:Gfp.

**Fig 3. Characterization of the Hog1-associated cytoplasmic foci in response to FA treatment, after Co-IP.** Foci content was subjected to Co-IP with GFP-Trap magnetic beads. Proteins significantly enriched in FA foci (P-value < 0.05, log2 fold-change > 1) are marked in orange. Puf2 (UniProt ID: N4WW71) in red, and RNA recognition motif (RRM)-containing proteins in blue. **A)** Volcano plot comparing proteins enriched in Hog1 foci to the DMSO control. **B)** Volcano plot comparing proteins enriched in Hog1 foci to untagged control. **C)** Venn diagram showing the overlap of significant proteins enriched in FA foci compared to DMSO (red circle) and untagged controls (blue circle). **D)** Sankey plot of GO terms enriched in the 90 shared proteins identified in FA foci. GO terms (Molecular Function, Cellular Component, Biological Process) were analyzed using DAVID (v2023q2). Circle size represents the number of enriched proteins, with Puf2 highlighted by an arrowhead. Fold enrichment is shown on the x-axis. **E)** Multi-panel scatter plots visualizing proteomic differences across experimental conditions. Each panel represents pairwise comparisons of proteomic differences between: Total proteome Co-IP, SGEF Co-IP (FA-DM), SGEF Co-IP (Hog1:GFP-Untagged C4). Points are colored according to the SGEF log2 FC values using an orange-to-gray-to-purple gradient, where orange represents the lowest values and purple the highest. Highlighted proteins, Puf2 and eIF2 (UniProt ID: N4WKP0), are marked.

The abundance of Puf2, an RNA-binding, pumilio-domain SG-associated protein was strongly FA-dependent (S2 Table, S6B and S6C Fig). A predicted *C. heterostrophus* ortholog of yeast Puf2 was identified by BLASTP searches and expressed as a C-terminal fusion to tdTomato. The C-terminal tdTomato Puf2 fusion protein localized to foci upon exposure to FA. In live imaging, starting about 30–40 min after application of FA the Puf2:tdT fluorescence signal increased and localized to previously formed Hog1:GFP-containing cytoplasmic foci (Figs 4C and S6B and S1 and S2 Videos). The simplest interpretation is that upon exposure to FA, Hog1:GFP localizes to SG or another LLPS structure which then recruits Puf2 or fuses with Puf2-containing SG. Formation of foci containing a known SG protein, Puf2 (16), is consistent with FA being a stress signal. In agreement with the time course (S6B Fig) which shows Hog1:GFP foci appearing before formation and colocalization of Puf2 foci, deletion of Puf2 did not prevent formation of Hog1:GFP – labelled foci, implying that Puf2 is not essential for the sequestering of Hog1 (S4 Fig). Although foci were present in the *puf2* deletion, more Hog1:GFP is apparent in the nuclei, alongside the sequestration to granules typical of WT cells (S4B Fig). Occasionally hyphae showing this pattern are seen in the parent strain (Hog1:Gfp in WT background) so this is not an absolute puf2 phenotype but may belong on a regulatory gradient between nuclear accumulation and sequestering to granules.

Co-immunoprecipitation (Co-IP) by Hog1:GFP also enriched for mitochondrial proteins, suggesting an association between ChHog1 and mitochondria under FA-induced stress. To follow the relationship between Hog1:GFP and mitochondrial dynamics upon exposure to FA stress, cells expressing an ATP1:mCherry fusion were exposed to 2.5 mM FA. This treatment resulted in rapid mitochondrial fragmentation. In DMSO-treated controls, mitochondrial fragmentation was not observed even after 30 minutes incubation. Hog1:GFP colocalized only rarely with fragmented mitochondria (S7 Fig, representative of 4 biological repeats). To assess a possible relation of the Hog1:GFP foci to peroxisomes, a mCherry protein whose C-terminus comprises the peroxisomal targeting signal SKL was expressed under the constitutive fungal Pgpd1 promotor. Spherical organelles larger than the Hog1:GFP-labelled foci and much less numerous were observed in a pattern distinct from the Hog1:GFP foci (S8 Fig; representative of 3 independent SKL transformants). Thus ChHog1 is not sequestered to peroxisomes.

### Relationship between dephosphorylation and foci formation

We then asked whether ChHog1 dephosphorylation upon exposure of the cells to FA occurs before or after incorporation into cytoplasmic foci. Immunofluorescence images of fixed cells probed with antibody to dual phosphorylated ChHog1 show a basal level of cytoplasmic and nuclear localization. The basal activation level, seen also on immunoblots, and basal level of Hog1 nuclear localization might result from light activation of Hog1, as reported for *Aspergillus* [19] as all our experiments were done under ambient light. Alternatively, the basal level could be a fundamental feature of MAPK signaling in fungi. Osmotic stress strongly increased dual phosphorylation combined with nuclear retention as shown by strongly fluorescent nuclei, particularly in appressoria (Figs 5 and S2). The basal level of dually phosphorylated Hog1 staining was not affected by exposure to DMSO solvent or CA as controls but was reduced by FA (Fig 5C). Dually phosphorylated ChHog1 at the locations of the Hog1:GFP foci was not detectable by immunofluorescence, although this signal

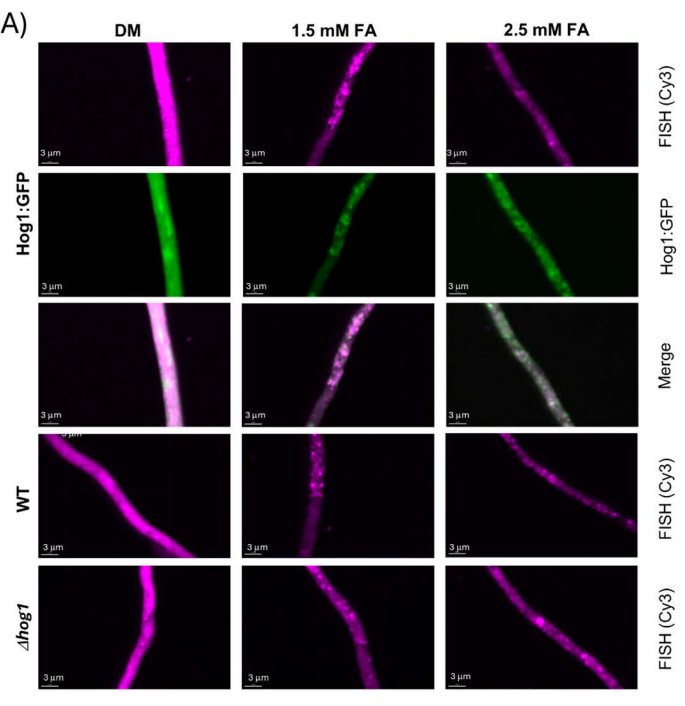

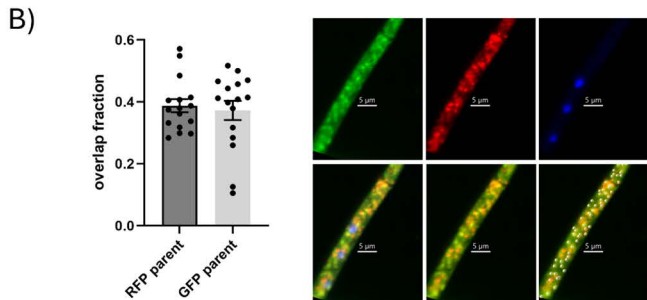

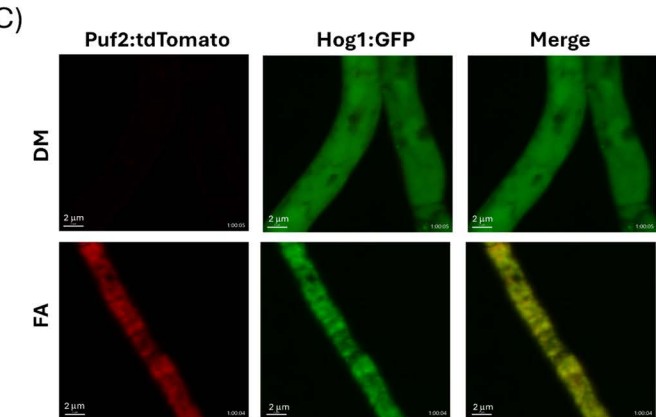

**Fig 4. Correlation of Hog1:GFP foci with mRNA-containing granules and the SG marker Puf2. A)** Detection of mRNA. *C. heterostrophus* cells expressing Hog1:GFP fusion protein were grown and prepared as for Fig 1 and exposed to DMSO (control), 1.5 mM FA or 2.5 mM FA. Cells were fixed with 4% paraformaldehyde, permeabilized, probed with oligodT:Cy3 (smFISH; see Methods) and imaged (Spinning disk confocal, x100). The magenta

channel is the FISH hybridization signal (Cy3), and the green channel is Hog1:GFP fluorescence. Δ*hog1*, Hog1 deletion in the C4 background. Images are representative of three hybridization sets (see **B**). Scale bars = 3 μm. **B)** Quantitation of overlap of Hog1:GFP and smFISH fluorescence signals. Spots were counted by Imaris software (for details see Methods) on 5-6 images from each of three hybridization sets on independent cultures (16 images total) and analysed for the number of smFISH signal (denoted parent, red) spots overlapping, containing or coinciding with Hog1:GFP spots, and inversely, the number of Hog1:GFP spots (denoted parent, green) overlapping, containing or coinciding with smFISH spots. Left, fraction of overlapping spots for each analysis direction plotted separately ("parent" denotes which channel was counted first). The fraction of overlapping spots among the total number of spots counted, calculated from the second analysis step and therefore independent of the direction of analysis, was 0.36 ± 0.07, mean and SEM of 16 images. Right panel, illustration of the spot analysis on a portion of a single image. First row: from left to right, green channel, Hog1:Gfp; red, smFISH; blue, DAPI. Second row: from left to right, all three channels; overlay, green and red channels; green and red channels, with the detected overlaps plotted as bright spheres. Scale bars 5 mm. **C)** Colocalization of the SG marker Puf2 with Hog1:Gfp foci. *C. heterostrophus* cells expressing both Hog1:GFP and Puf2:dTtomato fusion proteins were prepared and treated as for Fig 1 except that the samples were not fixed, and live images were collected at 10 min intervals over 1 h. An image of Hog1:GFP and Puf2:dTtomato signals and their overlap at 60 min post FA induction is shown; representative of two independent experiments. Scale bars, 2 μm. For animations of the full time series see S1 Video (DM) and 2 (FA). Hog1:GFP foci (green channel) are clearly visible by 20 min and Puf2 foci (red channel) around 40 min.

is just visible in immunoblot assays of FA-treated samples (Fig 1E). One could propose that dephosphorylation is required for incorporation into foci. This was tested by FA treatment of a PtcB deletion strain that expresses the Hog1:GFP fusion protein (PtcB is a ChHog1 phosphatase [15]. Formation of foci was normal in the PtcB deletion strain *Dptcb* (S4 Fig), so FA-induced dephosphorylation, which is lacking when PtcB is deleted, is not a prerequisite for localization of ChHog1 to the foci. Likewise, lack of the phosphatase CDC14, also required for ChHog1 dephosphorylation [15] does not affect foci formation (S4 Fig). Immunoblot analysis showed that Hog1 is significantly dephosphorylated in the FA foci (SGEF) fraction as compared to the control (Fig 1E).

FRAP was used to monitor the mobility of HOG1:GFP under different conditions. The recovery from photobleaching depends on the mobile and immobile fractions of a protein population, based on their ability to diffuse back into the bleached region. In untreated hyphae of both WT and *Dptcb* it is clearly observed that HOG1:GFP is fully mobile. The protein is efficiently bleached in the region of interest (ROI), and fluorescence recovery reaches approximately 20% within 4 minutes, likely due to diffusion of HOG1:GFP from adjacent regions. This indicates the presence of a mobile fraction, though recovery might be limited by the size of the bleached region: in untreated hyphae bleaching extends beyond the focused region and usually covers an entire compartment delineated by septa. This was not mitigated by lowering the laser intensity (in the range needed for bleaching); the basis of this effect needs further investigation. Under stress treatment ChHog1:GFP is localized to either nuclei (osmotic stress) or granules (FA stress), and only the targeted regions are bleached as expected for FRAP experiments (S10 Fig). Under sorbitol treatment, we see rapid and substantial recovery in the nucleus, reaching ~50% recovery within 4 minutes. This suggests that cytosolic HOG1:GFP serves as a reservoir (buffer) that replenishes the nuclear pool after bleaching. In contrast, under FA treatment, there is clear evidence of HOG1:GFP immobilization. The bleached ROI remains sharply defined, and there is no observable fluorescence recovery over time. This indicates the absence of a mobile fraction capable of diffusing into the bleached area. These data are in agreement with the known shuttling of Hog1 between nucleus and cytoplasm under osmotic stress, as well as with the stability of Hog1:GFP fluorescent granules which show only partial recovery even after several hours incubation (S2 Fig). Finally, no significant differences were observed between WT and *Dptcb* regarding their response to photobleaching in any of the tested conditions (S10 Fig), providing further genetic evidence that dephosphorylation is not needed for sequestering.

## Relationship between responses to osmotic stress and FA

Sorbitol at 1 M inhibits growth, though the wild-type fungus is able to grow under this osmotic stress. Addition of increasing concentrations of FA at sub-lethal levels suppresses the ability of the fungus to tolerate osmotic stress, as shown in a checkerboard growth assay (Fig 5). FA also prevents nuclear accumulation of Hog1:Gfp upon osmotic stress (Figs 5B, S3A and S3B). Furthermore, FA prevents expression of a reporter of osmotic stress, the monosaccharide transporter gene ChMST1 [8] (Fig 5D). Together, these results imply that FA suppresses the response of the fungal cell to osmotic stress.

## Discussion

Exposure of the fungal pathogen to the plant defense compound FA promotes formation of mRNA-containing granules, and sequesters ChHog1:GFP to cytoplasmic granules. These two intracellular processes are overlapping but not identical. Sequestering of ChHog1 to the fungal cytoplasm would directly prevent nuclear accumulation and consequent activation of downstream gene expression. Suppression of dual phosphorylation and nuclear accumulation would together suppress any phosphorylation-dependent activity of ChHog1 in interaction with cytoplasmic and nuclear target proteins. These mechanisms could impact the host-pathogen interaction in several ways. Sequestering could be a fungal survival response mitigating hyperactivation of the stress-dependent MAPK cascade. Alternatively or in parallel, the host may exploit the sequestering and dephosphorylation responses to dampen the pathogen's stress response.

### Nature of Hog1-containing foci

RNA FISH hybridization, proteomic and microscopy data are all consistent with identification of a subpopulation of the ChHog1-associated foci as stress granules (SG). These cytoplasmic mRNP granules have been proposed to intercept and sequester signaling components [20,21]. Exposure of *Candida boidinii*, *Pichia pastoris* and *Schizosaccharomyces pombe* to high-temperature stress resulted in colocalization of the P38 orthologs with the stress granule marker Pab1 [22]. Under heat stress, the *S. pombe* Sty1 MAPK associated with SGs containing the ortholog of the pumilio domain protein Puf2 [17]. In other cases, Hog1/P38 orthologs were found to affect cellular processes linked to SG formation, though indirectly [23]. Here, ChHog1 was dispensable for granule formation (Fig 4A). SGs containing ScHog1 are observed when yeast cells begin to enter stationary [24]. In mammalian cell lines, sequestering of the signaling scaffold protein RACK1 to SGs under some stresses inhibits P38 MAPK-mediated apoptosis [25], precedent for our hypothesis that sequestering of ChHog1 can dampen its activity. P38 associates with SG in mammalian cells under heat stress [17]. LLPS condensates provide a platform to assemble a unique oncogenic Ras/MAPK signaling pathway [26], implying that the presence of signaling proteins in condensates is not only a means to sequester them, but can be employed by cells to compartmentalize and assemble signaling pathways. In the *Neurospora crassa* circadian clock, condensation of mRNA to LLPS provides direct control of translation condensates may play a conserved role in circadian clock regulation [27–30].

Proteomic analysis showed significant enrichment of RNA binding proteins and translation initiation factors in Hog1:GFP foci (Fig 3D). The yeast orthologs of two of the RRM domain RNA binding proteins, MRN1 and NRP1, were found in the SG proteome in two previous studies [16,18]. Moreover, Hog1:GFP foci content is significantly enriched for 12 proteins (including the above two) previously associated with SGs. Puf2, a known SG-associated RNA-binding protein, is enriched following FA treatment (Fig 4C, S2 Table). Puf2 is a translational repressor [31]. Thus, Puf2 could repress translation of specific mRNAs associated with Hog1 FA stress-dependent condensates during its extensive overlap with Hog1 in a specific time window (Fig 4). To date, the best-understood mechanism for SG formation involves the stress-induced phosphorylation of the translation initiation factor 2 alpha subunit (eIF2α), which subsequently hampers translation initiation [32]. Notably, Puf2 and eIF2α(N4X6B8) were both enriched in all SGEF and Co-IP experiments. Based on colocalization, proteomic data and sm-FISH, we propose that a subpopulation of ChHog1 is sequestered in stress granules or in complexes that dock to stress granules, upon exposure to the plant antimicrobial compound, FA.

### Damping of Hog1 activation

FA is abundant in the leaf tissues of the host plant of *C. heterostrophus*, and acts on the pathogen as a stress, causing rapid cell death with hallmarks of apoptotic-like RCD [3]. Although this remains to be tested *in planta*, high levels of FA in the maize host cell walls and leaf cuticle could contribute to the (paradoxical) extensive cell death observed in necrotrophic fungal pathogens like *C. heterostrophus* during their initial attack on the plant host, when one might have expected maximal metabolism and growth needed to support invasion [33]. Stressors generally activate MAPK P38 orthologs like

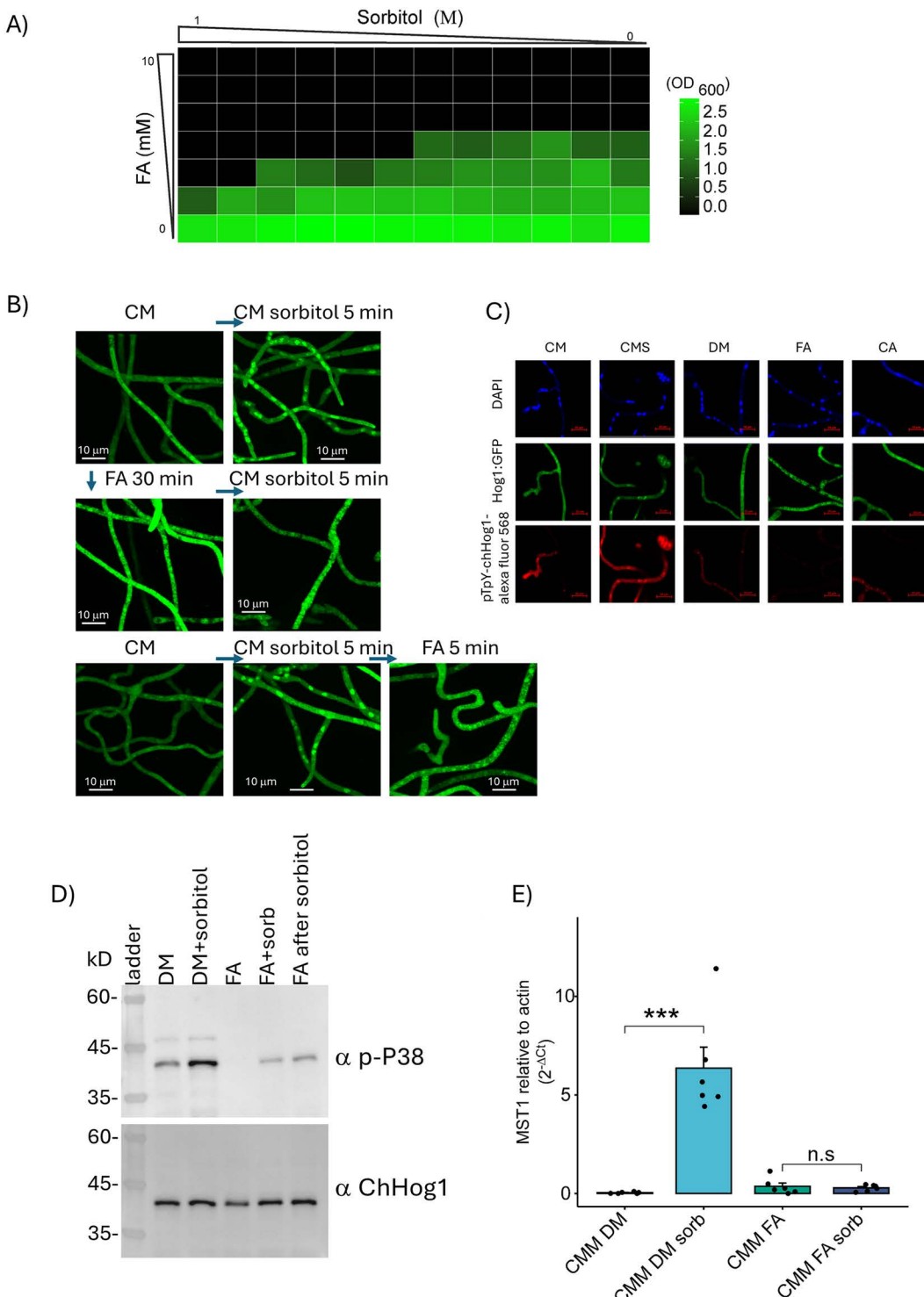

Fig 5. Interaction between osmotic stress and FA. A) Checkerboard growth assay. Growth at combinations of increasing sorbitol and FA concentrations (x and y axes) is plotted as a heat map. Average of two independent experiments, each with 25 technical repeats. B) FA-induced sequestering over-rides nuclear localization in response to osmotic stress. Expanded portions of images from S3A and S3B Fig are shown here. Preincubation with FA prevents nuclear accumulation of Hog1:Gfp upon exposure to 1 M sorbitol, with the Gfp signal remaining, instead, in cytoplasmic puncta. C) Subcellular mapping of the phosphorylation state of ChHog1, under stress treatments. Cultures were grown and treated as for Fig 1, exposed to osmotic stress,

 

FA, caffeic acid (CA) or heat shock, fixed, and subjected to immunohistochemistry (see Methods). Dually phosphorylated Hog1 was detected by ap-P38 antibody (Cell Signaling, T180-Y182, 9211S) (1st Ab 1:400, 1 hr) followed by anti-rabbit Alexa Fluor 568 (2nd Ab 1:1000, 1 hr). Mounting was with DAPI Fluoromount-G (SouthernBiotech CAT NO 0100-20) and samples imaged with a Zeiss LSM 700 confocal microscope. CA, 5 mM caffeic acid. Scale bars = 10 mm. Images are representative of 3 biological repeats. **D)** Expression of osmotic-stress dependent transcript MST1. Stationary cultures (see Methods) were exposed to 1 M sorbitol, 2.5 mM FA, or both stresses, as follows: 15 min pretreatment with CMM (CM with glucose replaced by maltose, see (7), with DMSO at the same concentration as for 2.5 mM FA, followed by 30 min incubation in CMM with DMSO or 2.5 mM FA, and with/without 1 M sorbitol. qPCR data are for a total of 6 cultures for each of the four treatments, from three independent experiments with two cultures in each treatment. Statistical significance by ANOVA followed by a Kruskal-Wallis test, in GraphPad Prism. **E)** Hog1 phosphorylation levels in response to osmotic stress and FA. Cultures were grown as for Fig 1, preincubated with fresh CM with DMSO (DM) for 15 min, followed by 15 min with DM, DM + 1 M sorbitol, 2.5 mM FA, and 2.5 mM FA + 1M sorbitol. Lane 5, 15 min DM + 1 M sorbitol, followed by 15 min 2.5 mM FA. Two identical gels were run in parallel, and the immmunoblots probed with anti-phospho-P38 (left) and anti ChHog1 (right). Representative of two independent experiments.

ChHog1 by dual phosphorylation followed by nuclear import. We found here that FA, instead, sequesters ChHog1 to cytoplasmic foci. Some fungal P38 orthologs like the one studied here have significant activation levels in the absence of stress; this may be a more general feature than sometimes assumed. Dephosphorylation, which could be a signaling event, is developmentally regulated, for example, in the early stages of *Aspergillus nidulans* spore germination [34]. Recent work on the *Candida albicans* CaHog1 pathway showed that in contrast to high osmotic stress which acts via the canonical MAPK cascade, oxidative stress promotes phosphorylation by inhibiting dephosphorylation, superimposed on basal Ssk1-driven Hog1 phosphorylation [2]. A normal level of ChHog1 is needed for tolerance of FA stress [3]. Optimal survival would thus require tuning of the ChHog1 activation level. Hyperactivation of fungal Hog1 is the main route to cell death triggered by the fungicide fludioxonil [12] and overexpression of the protein phosphatase MoPtp2 in the rice blast pathogen *Magnaporthe oryzae* lowers MoHog1 phosphorylation, counteracting cell death [35]. Dephosphorylation of ChHog1 would, in this model, act as a response by the cell to mitigate the cell-death-promoting activity of FA (which is not, in itself, a consequence of ChHog1 activation). Growth rate analysis of the protein phosphatase-lacking mutant *ptcB* provided evidence that dephosphorylation is needed for WT tolerance of FA stress [15].

Another sensitive mechanism to dampen ChHog1 activity and prevent hyperactivation of the pathway could be its sequestering to a compartment where it cannot interact with downstream cytoplasmic or nuclear partners. Given that FA lowers ChHog1 phosphorylation, and acts inversely to fludioxonil [3], sequestering may not play a central role in the survival response to FA as single stressor *in vitro*. It is likely, though, to be a factor in the multi-stress environment on the plant host. Multiple stresses, some raising and some lowering ChHog1 dual phosphorylation, determine the phosphorylation state. Illustrating this point for our system, after or during exposure to FA, osmotic stress increases phosphorylation ( [15] and Fig 5D), with the two stresses acting in an opposite, and apparently additive, manner. FA strongly reduced the extent of hyperphosphorylation in response to osmotic stress (Fig 5D). Sequestration to foci is maintained under these different conditions, is functionally dominant to nuclear accumulation under osmotic stress (Figs 5 and S3) and represents an immobile fraction of the Hog1 population (S10 Fig).

## Hog1 sequestering and its possible roles in the host-pathogen interaction

If sequestering of ChHog1 acts to dampen its hyperactivation under stress, this raises the question of how sequestering and dephosphorylation are integrated. The FA signal is dominant (in the biochemical sense) to osmotic stress: exposure to sorbitol could not release Hog1 from the granules, while after exposure to FA, ChHog1 exited the nucleus and accumulated in granules, even under osmotic stress. Furthermore, ChHog1 is sequestered primarily in its dephosphorylated state (Figs 1E and 5C). Dephosphorylation of activated ChHog1 could, in principle, act as a signal, even though dual-phosphorylated MAPKs are normally the active signaling species. Dephosphorylation could then trigger incorporation into SG. Genetic evidence does not favor this, and it seems more plausible that distinct signaling hierarchies lead to granule formation and ChHog1 sequestering. Indeed, mutants lacking Hog1 phosphatases PtcB or CDC14 still sequester ChHog1 (S4 Fig). ChHog1 is not required for mRNPs granules formation (Fig 4A), as reported previously for the *S. pombe*

ortholog [18]. The RNA-binding protein Puf2, although itself sequestered upon FA treatment, is dispensible for formation of Hog1:GFP foci (S4 Fig), although some of the images suggest a partial loss of regulation leading to more frequent presence in nuclei under FA stress, which needs to be investigated further.

FA induced rapid mitochondrial fragmentation within 10 min, as reported in *Aspergillus fumigatus* following oxidative stress [36]. The pattern of Hog1:GFP foci is distinct from fragmented mitochondria; both fluorescence signals are abundantly distributed but there are only few instances of overlap, at the later time points (S7 Fig). The dynamic nature of LLPS and the observation that they can contain key components of signaling pathways, forms a cyclic relationship in which SG or other LLPS can alter signaling and post translational modification which in turn influence SG assembly [20,37,38]. Conversely, formation or disassembly of stress-induced condensates may be regulated by kinases as in the case of Grb7 phosphorylation in carcinoma cells [37]. In *Schizosaccharomyces pombe*, the NDR/LATS kinase Orb6 phosphorylates the RNA-binding protein STS5 under non-stress conditions. Dephosphorylation of STS5 under stress conditions then permits stress granule formation [39,40].

Sequestration of ChHog1, preventing its accumulation in the nucleus, may interfere with stress signaling. Seen from this angle, FA as a host defense factor could prevent an efficient stress response in the fungus. In general, Hog1/P38 orthologs in fungi are involved in tolerance of osmotic and oxidative stresses; and as we have shown, to FA itself. Testing the hypothesis that FA suppresses the stress response of the pathogen, we found that FA prevents nuclear accumulation of Hog1:Gfp in response to osmotic stress, hyperphosphorylation, and expression of the downstream target gene MST1. MST1 transcript levels are very low in the absence of Hog1 or its activation by stress ( [8]; Fig 5). FA at 2.5 mM which saturates cytoplasmic sequestering of Hog1, also prevented MST1 expression in response to osmotic stress (Fig 5), suggesting that this plant host signal interferes with stress signaling. This mechanism could have profound implications for the plant-fungal interaction and might also provide a future strategy to block stress signaling by pathogens. Despite the central roles of Hog1 orthologs in fungal pathogens, it is not clear to what extent they are exposed to osmotic stress in their hosts, which also produce oxidants as well as known and uncharacterized defense compounds. The full nature and variety of stresses need to be studied in the interaction of fungal pathogens with plant and other hosts.

Independently of Hog1:Gfp and its sequestering, the FISH localization of mRNA and the formation of Puf2:dtTomato foci together imply that FA is a stress signal that (like heat shock, for example) leads to LLPS condensation. In a model of plant interspecific competition, plant phenolics interacted directly with RNA binding protein RBP47B, promoting its phase separation and triggering SG formation [41]. This could hint at the mechanism in fungi as well. Given the key role of MAPK import to the nucleus in eukaryotic cells in general, we expect that signal-dependent sequestration of the stress-activated MAPK, as found here, will have implications well beyond the fungal kingdom, in normal developmental cascades and in disease.

## Materials and methods

### Generation of transgenic fungal strains

Strains used in this study and oligonucleotide sequences are given in S1 Table. Deletions were made using either the split marker protocol [42] or *via* transformation with a plasmid derived from pUCATPH [43,44] with the 5' and 3' flanks of the gene of interest flanking the hygromycin phosphotransferase (Hyg) expression cassette. The amino acid sequence of the ScPUF2 RNA-binding protein was retrieved from the Uniprot database (http://www.Uniprot.org/) and *C. heterostrophus* homologs of these genes were identified using BLASTP searches of the JGI database Home - Cochliobolus heterostrophus C4 v6.0 (doe.gov) [45]. DNA fragments of these genes were amplified using the oligonucleotides given in S1 Table. GFP tdTomato and mCherry tagging protein sequences were retrieved from Addgene (https://www.addgene.org/), optimized for *C. heterostrophus* codon usage using the IDT (https://eu.idtdna.com) codon optimization tool according to the genome of the related Dothideomycete *Alternaria alternata*, and synthesized (IDT, https://eu.idtdna.com) with

appropriate restriction sites. Bait protein and tag sequences were restriction digested and ligated into pNG1 containing a neomycin/G418 resistance cassette (kindly provided by Prof. B. Gillian Turgeon, Cornell University) (for the ChHog1:GFP protein) or into pUCATPH containing a hygromycin B resistance cassette (for the ChPuf2::tdTomato protein). A Pgpd1:m-Cherry:SKL fusion protein as a genetically encoded peroxisome marker was expressed from a synthesized sequence. HGP::ATP1:mCherry strain was generated using NEBuilder HiFi DNA Assembly kit, with pUCATPH plasmid as template and primers as detailed in S1 Table.

For transformation of *C. heterostrophus*, plasmid DNA was linearized and introduced by the PEG–Ca$^{++}$ method [46]. To construct the HGP513 strain, *C. heterostrophus* race T strain C4 from the Cornell University collection was grown on CMX (complete medium, CM, with glucose replaced by D-xylose, see [46] at 22°C for two weeks in a 12 h light/12 h dark regime. These plates were used for preparation of protoplasts. Spores were inoculated to 150 mL CM and cultured in a rotary shaker for 18 h at 30°C, 200 rpm. The fungal suspension was centrifuged at 8,000 rpm for 10 min (Sorvall, SLA1500 rotor). The resulting germling pellet was suspended in 70 mL of enzyme-osmoticum solution, incubated 2.5 h at 30°C, 70 rpm, and protoplasts filtered through gauze. The filtrate was centrifuged at 4500 rpm for 5 min at 4°C and proto-plast pellets were washed with 10 mL cold 0.7 M NaCl, followed by a second wash in 10 mL STC (sorbitol-Tris-Ca$^{++}$ buffer, [46]. The resulting protoplasts were suspended in 200 µl of STC, counted and adjusted to a concentration of $10^8$ cells/mL. $10^7$ protoplasts were used for each transformation. The protoplasts were incubated on ice with 25 µg of linear transfor-mation construct DNA for 12 min (controls without DNA were subjected to the same procedure). 60% w/v MW 3,350 PEG (Sigma) was added in three aliquots of 200, 200, and 800 µL each and then diluted with 1 mL STC and plated for regener-ation. A 1% agar overlay containing 1 mg/mL G418 (Gibco) (Neo) was added after 18 h for selection. One control plate to test viability was overlayed with agar without Neo. All additional strains were established starting with the HGP513 strain using hygromycin B (HYG) 100 µg/mL as a secondary selection [46]. For single spore isolation, newly formed colonies after transformation were transferred to 10 mL CMX-Neo 1 mg/mL (for the HGP513 strain) or CMX-Neo 1mg/mL and Hyg 100 µg/mL (for all additional strains) plates. Mature spores were collected, diluted in sterile water so that up to 10 spores were dispersed on a 25 mL CMX-Neo 1 mg/mL (for the HGP513 strain) or CMX-Neo 1 mg/mL – Hyg 100 µg/mL (for all additional strains) plates allowing new colonies to be formed - each from a single spore.

## Verification of gene replacements

The replacements of genes by the fusion constructs (for the HGP513; HGP513-PUF2-tdTomato strains) were verified by PCR. Each fusion was tested with a primer upstream to the gene paired with a primer corresponding to the end of the tag (e.g., Hog1-Up Stream & GFP-EcoRI rv). The replacements of genes by the hygromycin resistance cassette (for the HGP513-ΔchPtcB, HGP513-ΔchCDC14 and HGP513-ΔchPuf2 strains) were verified by three PCR amplifications: first, with primers designed from the gene sequence (ORF) verifying the deletion of the gene, and second, with primers upstream and downstream of the gene's flanks used to construct the knockout mutant, and third, primers within the hygro-mycin phosphotransferase coding sequence.

## Stress induction

*C. heterostrophus* cells expressing the above fusion proteins were grown on selective plates as indicated for 2 weeks. Spores were collected with sterile water, filtered through sterile gauze and $2*10^5$ cells in 25 ml sterile water were incu-bated in a 150 mm plate, containing microscopy cover slips, for 5–6 h until spores attached to the surface and germinated. The water was replaced with complete medium (CM) and the plates incubated 16 hr. For osmotic stress the CM was replaced with fresh CM (control) or CMS (CM supplemented with 1 M sorbitol) for the indicated time at RT. FA was applied by replacing the CM with either DM (CM supplemented with the same volume of DMSO, the solvent for ferulic acid stock, as the FA volume added) or FA (CM supplemented with 2.5 mM ferulic acid) for the indicated time at RT. For heat shock (HS) the CM was replaced by CM preheated to 45°C and incubated for 10, 30 or 60 min at 45°C. After stress treatments,

cells were fixed in 4% paraformaldehyde in PBST for half an hour, washed with PBST and imaged. Live imaging was also performed for both GFP and tdTomato constructs with 200 cells in 100 µl medium in a 8 or 18-well u-Slide (ibidi, Gräfelfing, Germany).

### Immunohistochemistry, FISH assay and confocal fluorescence microscopy

Cells were induced and fixed as indicated above and then rinsed three times with PBST and their cell wall was permeabilized by digestion with enzyme solution consisting of 67.5 mg/ml glucanase (VinoTaste, Novo, Denmark), 22 mg/ml Driselase (Sigma) and 2 mg/ml chitinase (Sigma) in PBS, on ice for 15 min. Cells were rinsed 3 times with PBST and a second permeabilization was done by adding 200 µl 0.1% Triton X-100 in 0.1% sodium citrate onto the slide, for 2 min. Cells were rinsed again and blocked with 3% BSA in PBST. Dually phosphorylated ChHog1 was detected by p-p38 antibody (Cell Signaling, T180-Y182, 9211S) (1st Ab 1:400, 1 h) and anti-rabbit Alexa Fluor568 (2nd Ab 1:1000, 1 h). For the FISH staining, following permeabilization with Triton-citrate, cells were treated with 200 mM vanadyl ribonucleoside (40 min, 30°C) and washed with hybridization buffer containing 4 x SSC, 50% formamide, 10% dextran sulfate, 125 µg/µl *E. coli* tRNA, 500 µg/ml salmon sperm DNA, 1x Denhardt's solution and 0.4 U/µl RNAsin for 1 hr at 30°C. Cells were then incubated with PolyA-Cy3 probe overnight at 37°C. Mounting was done using DAPI Fluoromount-G (SouthernBiotech CAT NO 0100–20).

To visualize mitochondria in live native conditions, a strain expressing an Atp1:mCherry fusion was imaged. Samples were imaged by Spinning Disk confocal (Yokogawa CSU-W1) microscope (Nikon) with a CFI PLAN APOCHROMAT 100x OIL objective (N.A.-1.45). DAPI, FITC (GFP), Alexa Fluor 568 and iRFP720 signals were captured by exciting with laser lines: 405, 488, 561 and 640 nm, respectively. Emission signals were collected with a sCMOS camera (Photometrics, PRIME –BSI) with 95% QE by using NIS software. Images were collected in Z-stack mode with 0.2-0.4 µm steps through 8–15 µm. Deconvolution was done using NIS-elements for images of strains carrying GFP constructs only. For images of strains carrying both GFP and tdTomato the NIS Denoise AI algorithm was applied. Data presented were obtained with the same imaging settings namely laser power, light path and camera exposure time from at least three independent experiments, all of which showed similar effects. Further image analysis was done with Imaris (Oxford Instruments) software. For the quantification of overlap in Fig 4B, shared detection of Hog1:Gfp (green channel) and mRNA FISH (red channel) spots was defined by the following parameters: green first (green channel parent), estimated diameter 0.5 mm, no background subtraction, quality >3.9, intensity mean of the green channel >250, 2 classes (green and red channels); intensity mean of the red channel >800. Red first (red channel parent), estimated diameter 0.5 mm, with automatic background subtraction, quality >50, intensity mean of the red channel >200, 2 classes (green and red channels); intensity mean of the green channel >220. The next step employed "colocalize spots" analysis (this should strictly be referred to as "overlap") on the results of the above two selections, with threshold value distance <=0.5 mm. In Fig 4B, the bar graph shows the result of the first analysis step. The mean number of overlapping spots given in the legend to Fig 4 is the result of the second analysis step.

Fluorescence recovery after photobleaching (FRAP) assays were done in a LSM 710 laser scanning inverted confocal microscope: the bleaching laser pulse and subsequent time course of the GFP fluorescence signal integrated over the region of interest (ROI) were controlled by and recorded with the Zeiss Zen software. A non-bleached ROI in each image was also marked and used as a reference to correct for overall decrease in signal as a result of the multiple scans (S10 Fig).

### Preparation of an SG-enriched fraction from induced samples and confocal fluorescence microscopy

Cultures were grown as described above in 150 mm plates. Exposure to 2.5 mM FA or DMSO (solvent-only control) was performed as described above. Cells were scraped immediately, and flash frozen in liquid nitrogen. Native protein extraction was performed using native buffer: 50mM Tris HCl pH 7.4, 100 mM potassium acetate, 2 mM magnesium acetate, 0.5 mM DTT, 0.5% NP40, 0.5 mM EDTA, 50 mM NaF, 50 mM β glycerophosphate, 1 mM sodium orthovanadate, 10

mM sodium pyrophosphate, aprotinin (Sigma A6279) 1:500, 0.005 mM Pepstatin A (P5318), 5 mM Leupeptin (L2884), 5 mM 110-Phenanthroline (131377), as described previously (16) with slight modifications. Briefly, 800 mg cells were ground in liquid nitrogen and then suspended in cold native buffer in a 1.5:1 ml:mg tissue ratio, vortexed vigorously for 1 min and let sit on ice for 3 min, repeating 3 times. Crude SG fraction was isolated by sequential centrifugation: nuclear fraction and debris were removed by low-speed centrifugation1500 x g, 4°C, 5 min. The supernatant was centrifuged at 18000 xg, 4°C, 15' after which the pellet contained the crude SG fraction. The total SG fraction was resuspended in 100 µl of native buffer and cleared again from residual aggregates by centrifugation at 1000 x g, 4°C, 2 min. Supernatant containing cleared SG was collected for further analysis by confocal microscopy, proteomic analysis and immunoblot analysis (Figs 2B and 1C). Confocal microscopy was as described for immunocytochemistry above. Quantification of maximal intensity projection images was performed using IMARIS software (version 9.3.1; Oxford Instruments).

## SG fraction co-immunoprecipitation

The crude stress granule fraction was loaded on GFP-Trap_MB beads (ChromoTek, Planegg-Martinsried, Germany) and rotated at 4°C for 2 hr. Samples were washed three times with lysis buffer (50 mM Tris HCl pH 7.4, 100 mM potassium acetate, 2 mM magnesium acetate, 0.5 mM DTT, 0.5% NP40, 0.5 mM EDTA, 50 mM NaF, 50 mM β glycerophosphate, 1 mM sodium orthovanadate, 10 mM sodium pyrophosphate, aprotinin (Sigma A6279) 1:500, 5 mM pepstatin A (P5318), 5 mM leupeptin (L2884), 5 mM 110-phenanthroline (131377). Samples were eluted with 40 µl 2X Laemmli sample buffer.

## Proteolysis and mass spectrometry analysis

60 µl of the SG fraction samples was mixed with 15 µl of 5X Laemmli sample buffer and loaded on a 4–15% Mini-PROTEAN TGX Precast Protein Gel (BioRad, cat # 4561084). The proteins in the gel were reduced with 2.8 mM DTT (60°C for 30 min), modified with 8.8 mM iodoacetamide in 100 mM ammonium bicarbonate (in the dark, room temperature for 30 min) and digested in 10% acetonitrile and 10 mM ammonium bicarbonate with modified trypsin (Promega) at a 1:50 enzyme-to-substrate ratio, overnight at 37°C. Second digestion was performed with 1:100 enzyme-to-substrate ratio, 4 hr at 37°C. The resulting tryptic peptides were desalted using C18 tips (Glygen C18 TOP-TIP), dried, and re-suspended in 0.1% formic acid. First biological repeat, including 3 technical repeats, run on a Q Exactive mass spectrometer (Thermo) in a positive mode using repetitively full MS scan followed by collision induces dissociation (HCD) of the 10 most dominant ions selected from the first MS scan. The peptides were resolved by reverse-phase chromatography on 0.075 X 300-mm fused silica capillaries (J&W) packed with Reprosil reversed phase material (Dr Maisch GmbH, Germany). The peptides were eluted with linear 120 minutes gradient of 5–28% 15 minutes gradient of 28–95% and 25 minutes at 95% acetonitrile with 0.1% formic acid in water at flow rates of 0.15 µl/min. Second biological repeat, including 3 technical repeats, run on a Q Exactive HF mass spectrometer (Thermo) in a positive mode using repetitively full MS scan followed by collision induces dissociation (HCD) of the 20 most dominant ions selected from the first MS scan. The peptides were resolved by reverse-phase chromatography on 0.075 X 300-mm fused silica capillaries (J&W) packed with Reprosil reversed phase material (Dr Maisch GmbH, Germany). The peptides were eluted with linear 120 minutes gradient of 6–30% 15 minutes gradient of 30–95% and 15 minutes at 95% (80% acetonitrile with 0.1% formic acid in water) at flow rates of 0.15 µl/min. The mass spectrometry proteomics data have been deposited to the ProteomeXchange Consortium via the PRIDE partner repository with the dataset identifier PXD044457. The mass spectrometry data from all repeats was analyzed using the MaxQuant software 2.1.3.0 (Mathias Mann's group) *vs* the *Cochliobolus heterostrophus* entry of the Uniprot database with 1% FDR (false discovery rate). The data were quantified by label free analysis using the same software. Statistical analysis of the identification and quantitation of results was done using Perseus 2.0.6.0 software (Mathias Mann's group). Statistically differential proteins were defined as: at least two peptides,with 2 sided t-test p-value < 0.05, log fold change > 1.

## Immunoblots

Frozen mycelium of induced samples was ground in liquid nitrogen, suspended in 5% SDS, 100 mM Tris pH 8 and 10 mM DTT, heated for 5' at 95°C and sonicated (Microson XL-Misonic, 30s, level 5). The extract was centrifuged (10000 xg, 10', RT) and the supernatant acetone-precipitated. The resolved protein pellet was solubilized by urea buffer containing 8 M urea, 100 mM ammonium bicarbonate and 10 mM DTT. Protein concentration was determined using Bradford assay and 30µg of each extract was loaded on a 10% SDS-PAGE. For the SG fraction, 20 µl mixed with 5X Laemmli sample buffer were loaded on 10% SDS-PAGE. Transfer was performed using Trans Blot Turbo protein transfer kit & system (BioRad). Blots were probed with either custom produced ChHog1 antibody (https://www.biomatik.com/) or p-p38 antibody (Cell Signaling, T180-Y182, 9211S) and imaged using Fusion Pulse (VILBER).

## Real-time PCR

Stationary cultures of wild type C4 were assayed 20 h after plating 5 10^4 conidia in 10 ml liquid CM in standard 9 cm petri plates. Medium was removed replaced with fresh CMM (as CM, with glucose replaced by maltose) medium with 2.5 mM FA or an equivalent amount of DMSO as solvent control. After 15 incubation the medium was replaced again, with the same media, with or without 1 M sorbitol, incubated for a further 30. The medium was removed by pouring off then standing the plates vertically for 3 min, the samples harvested with a cell scraper, and immediately frozen in liquid $N_2$. Samples were ground to a fine powder in liquid $N_2$, extracted by adding an approximately equal volume of TriReagent (Sigma) to the frozen powder. Samples were vortexed 20 s, centrifuged 1 min at 22,000xg to remove cellular debris, and purified on Zymo Direct-zol columns following the manufacturer's protocol including on-column DNA digestion. 1 mg total RNA was used for cDNA synthesis (Quanta qScript) in a 20 ml reaction, the products diluted 6-fold in nuclease-free water, and 1.5 ml/well used for real-time PCR amplification (QuantStudio, Applied Biosystems) in a 15 ml reaction with SYBR reaction mix (Quanta PerfeCTa SYBR Green FastMix) and the following primer pairs (IDT; designed from sequences downloaded from the *C. heterostrophus* C5 v6.0 database at Mycocosm, Joint Genome Institute (46, 47), with IDT's online tool, with one member of the pair crossing an exon-intron boundary) actin (JGI Mycocosm ID 15010, forward: TCACCATGGTATCATGATTGGTAT reverse: TGTCATCCCAGTTGGTAACG) and MST1 (JGI Mycocosm ID 211578; NCBI ABY48860, reference 6; forward: AGGCTGAGTGCTGCAAG; reverse: GAGACAACCATAGTCGATCCAG). Control reactions with actin primers on templates prepared without reverse transcriptase gave Ct values of higher by about 15 relative to similar parallel cDNA samples indicating negligible genomic DNA contamination. cDNA samples from non-induced cultures also yielded amplification with the MST1 primer pair that was near or below the limit of detection.

## Growth assay and kinetics measurement

*Cochliobolus heterostrophus* strain C4 was cultured in a 96-well microtiter plate (150 spores/well) containing a two-dimensional matrix of ferulic acid (FA) and sorbitol. FA was serially diluted two-fold from 10 mM to 0 mM across rows A–G, and sorbitol was diluted two-fold from 1.0 M to 0 M across columns 1–12. Each well contained 200 µL of complete medium supplemented with the appropriate combination of FA and sorbitol. Plates were incubated at 24–28°C without shaking.

Fungal growth was monitored using a Synergy S1 plate reader (BioTek) configured for discontinuous kinetic reads over a 96-hour time course. Absorbance at 600 nm ($OD_{600}$) was measured every two hours (49 total time points). Each well was scanned using a 5×5 point matrix (25 measurement points), evenly spaced across the well diameter, and the values were averaged per well per time point.

## Supporting information

**S1 Fig. Functional validation of Hog1:GFP fusion protein strain (HGP513) in comparison to the WT, and additional information relevant to** Fig 1**.** A. Growth rate. *C. heterostrophus* WT strain C4 and cells expressing Hog1:GFP fusion protein (HGP513) were grown on selective plates as indicated: CM (control), CMS (CM

supplemented with 1 M sorbitol for osmotic stress), DM (CM supplemented with DMSO, the ferulic acid solvent) or FA (CM supplemented with DMSO and 2 mM ferulic acid). Colony diameter was measured for 8 days. B. Virulence on maize leaves. Both strains were inoculated by agar blocks on 3-week-old maize plants and incubated for 96 h in a clear plastic bag to provide 100% relative humidity. Lesion areas were measured. Error bars indicate standard error of the mean for 12 lesions on a total of 6 leaves. C. Immunoblot for ChHog1 phosphorylation. Cells expressing Hog1:GFP fusion protein were grown and treated as for Fig 1A. Samples were collected by scraping from the plates immediately, or after the indicated times. Proteins extracted were subjected to immunoblot analysis using anti phospho P38 antibody or anti total ChHog1 antibody. D. Constitutive free Gfp localization. *C. heterostrophus* cells expressing PGPD1:GFP (GFP under the *Aspergillus nidulans* Pgpd constitutive promoter) were grown as for Fig 1A. Sampling was at 10 min (for the CM and CMS treatments) and 30 min (for the DM and FA treatments). E. Dependence of foci formation on FA concentration. *C. heterostrophus* cells expressing Hog1:GFP fusion protein were exposed to FA as indicated (live imaging; spinning disk confocal, x100, scale bars 5 mm). The bottom two expanded panels show more examples for 1.5 mM FA, near the threshold for foci formation, where Hog1:Gfp foci are often formed in a hyphal cellular compartment while the neighboring cell shows a uniform distribution or a basal level of nuclear localization.
(TIF)

**S2 Fig. Hog1:Gfp localization: timeline of response to osmotic stress, phenolic acids and heat shock.** *C. heterostrophus* cells expressing Hog1:GFP fusion protein were prepared, treated, fixed and imaged as for Fig 1. Sampling was at 10, 30 and 60 min after which induction media were removed, cells were washed once with CM and incubated for an additional 1 h in fresh CM (60 min recovery). GFP (green), DAPI (blue) and overlay are shown at three time points. Scale bars in each panel (expand for viewing on-screen) are 10 mm. At each time point three channels are shown for each treatment: GFP (green), DAPI (blue) and their overlay.
(TIF)

**S3 Fig. Sequential and combined induction.** *C. heterostrophus* cells expressing Hog1:GFP fusion protein were grown and prepared as for Fig 1. Scale bars (lower left of all panels; the Figure should be expanded to view on-screen, 10 mm). Representative of two biological repeats. Sequential induction (SI) was done as follows: A. SI CMS-FA – CM media was replaced with CMS (CM supplemented with 1 M sorbitol), incubated 5 min at RT, the medium was removed, and cells were exposed to either: DM (CM supplemented with DMSO) or FA (CM with 2.5 mM FA). B. SI FA-CMS – CM was replaced with FA, incubated 30 min at RT, the medium was removed, and cells were exposed to either CM or CMS.
(TIF)

**S4 Fig. Foci formation in WT Hog1:GFP and mutant strains.** *C. heterostrophus* cells expressing Hog1:GFP fusion protein were prepared as for Fig 1. Foci formation was imaged in the Hog1:Gfp parental strain and its deletion mutants in the genes encoding ChHog1 phosphatases (PtcB, CDC14) and Puf2 (predicted RNA-binding protein).
(TIF)

**S5 Fig. Additional proteomic data related to Fig 2.** A) Heat map of the total proteins identified in the SG fraction and proteins differentially elevated in the total SG FA fraction (3 biological repeats, 12 technical repeats, p-value < 0.05, log fold change > 1). B) Venn diagram of differential proteins elevated in the total SG FA fraction *vs*. Co-IP SG FA fraction from both tagged and untagged ChHog1. C) Scatter plot of bound protein in Hog1:GFP Co-IP *vs*. total SGs fraction. Orange dots indicate mRNAs significantly enriched (fold enrichment >2) in Hog1:GFP Co-IP. D) Immunoblot of total protein prior to SG fractionation from controls (DMSO) or FA-treated dHog1, C4 WT mycelia or Hog1:GFP strains; blots were probed (1st antibody) with anti ChHog1 or anti tubulin (see Methods) (3–4 biological repeats).
(TIF)

**S6 Fig. Additional information for** Fig 4**.** A. Detection of mRNA in granules: additional control images. The green channel images (Hog1:GFP) for C4 WT and *Dhog1* are shown. Rows 1 and 3 are reproduced here from Fig 4 for convenient comparison. Scale bars 1 mm. B. Overlap of the SG marker Puf2 with Hog1:Gfp foci. *C. heterostrophus* cells expressing both Hog1:GFP and Puf2:dTtomato fusion proteins were prepared and treated as for Fig 1 except that the samples were not fixed, and live images were collected at 10 min intervals over 1 h. Images representing selected time points are shown; for animations of the full time series see S1 Video (DM) and 2 (FA). Hog1:GFP foci (green channel) are clearly visible by 20 min and Puf2 foci (red channel) around 40 min. C. Quantitation of Puf2:dTtomato signal intensity during time. Total mean intensity was calculated from the images using Imaris software. Symbols indicate mean and SEM of 3 biological repeats, 6 fields, significant differences at 40–60 min by paired 2-tailed t test at $P < 0.005$. (TIF)

**S7 Fig. Comparison of Hog1:Gfp and mitochondrial distribution and dynamics.** *C. heterostrophus* cells expressing both Hog1:GFP & Atp1:mCherry fusion proteins, were grown and prepared as for Fig 1 without fixation, and were exposed to either DMSO (control, DM) or ferulic acid (FA) as indicated. Live imaging was done at 10, 30 and 60 min from the start of treatment. (TIF)

**S8 Fig. Comparison of Hog1:Gfp distribution with peroxisomes and nuclei.** *C. heterostrophus* cells expressing Hog1:GFP fusion protein and mCherry (under Pgpd1 promotor) comprising a SKL signal peptide at the C-terminus were prepared and imaged as for Fig 1. Scale bar, 10 mm. (TIF)

**S9 Fig. Subcellular fraction protein markers.** Protein Intensity values in the SGEF were normalized to protein intensity from whole proteome and FA to DM ratio was calculated. Values are normalized to the cytosolic housekeeping protein GAPDH (Cyto-G3PD). * p value <0.05; ** p value <0.01; *** p value <0.001. Nuc (Nucleus); Cyto (Cytoplasm); Per (Peroxisome); Mito (Mitochondria). (TIF)

**S10 Fig. Fluorescence recovery after photobleaching (FRAP) assays of ChHog1:Gfp mobility.** The central spot indicated by a red frame was bleached (middle panels). Neighboring framed spots were used to record non-bleached control signal levels. Images were recorded at 0.13 min intervals. In the graphs, full and empty symbols indicate two independent experiments, which were combined by normalization to the recovery at 3 min after the bleach, subtraction of the first recording after bleach, followed by scaling to the average of the total recovery in the two experiments. Graphs were generated in GraphPad Prism 10. (TIF)

**S1 Video. Colocalization of the SG marker Puf2 and Hog1:Gfp, control.** Time lapse record for the DMSO solvent control. The live image frames were recorded every 10 min after replacement of the medium. The last (60 min) image is shown in Fig 4B. *C. heterostrophus* cells expressing both Hog1:GFP (green channel) and Puf2:dTtomato (red channel) fusion proteins were prepared and treated as for Fig 1 except that the samples were not fixed. Scale bar is 5 µm. (MP4)

**S2 Video. Colocalization of the SG marker Puf2 and Hog1:Gfp, FA.** Time lapse record for the FA treatment. The live image frames were recorded every 10 min. The last (60 min) image is shown in Fig 4B. *C. heterostrophus* cells expressing both Hog1:GFP (green channel) and Puf2:dTtomato (red channel) fusion proteins were prepared and treated as for Fig 1 except that the samples were not fixed. Scale bar is 5 µm. (MP4)

**S1 Table. Lists of fungal strains and their genotypes, and oligonucleotides.**
(XLSX)

**S2 Table. Complete proteomic data.**
(XLSX)

## Acknowledgments

We thank Asmin Tulpule (Memorial Sloan Kettering Cancer Center) and Oded Kleifeld (Biology, Technion) for their insightful comments and suggestions. We are indebted to the Smoler Proteomic Center for their cooperation and support

## Author contributions

**Conceptualization:** Rina Zuchman, Ofri Levi, Benjamin A Horwitz.

**Data curation:** Rina Zuchman, Tamar Ziv, Ofri Levi.

**Formal analysis:** Tamar Ziv, Benjamin A Horwitz.

**Funding acquisition:** Benjamin A Horwitz.

**Investigation:** Rina Zuchman, Roni Koren, Tamar Ziv, Yael Lupu-Haber, Nitsan Dahan, Ofri Levi, Benjamin A Horwitz.

**Methodology:** Rina Zuchman, Roni Koren, Tamar Ziv, Yael Lupu-Haber, Nitsan Dahan.

**Project administration:** Benjamin A Horwitz.

**Resources:** Benjamin A Horwitz.

**Software:** Yael Lupu-Haber.

**Supervision:** Benjamin A Horwitz.

**Validation:** Roni Koren, Tamar Ziv, Ofri Levi, Benjamin A Horwitz.

**Visualization:** Benjamin A Horwitz.

**Writing – original draft:** Rina Zuchman, Benjamin A Horwitz.

**Writing – review & editing:** Ofri Levi, Benjamin A Horwitz.

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
