## [Decision Letter · Decision Letter 0]

26 May 2025

Cytoplasmic sequestering of a fungal stress-activated MAPK in response to a host plant phenolic acid

PLOS Pathogens

Dear Dr. Benjamin A Horwitz,

Thank you for submitting your manuscript to PLOS Pathogens. Your manuscript has undergone a full evaluation at the editorial level, along with rigorous peer review by three independent experts. While the reviewers acknowledged the significance of the problem you addressed, they raised several substantial concerns regarding the current version of the manuscript. These issues must be thoroughly addressed before we can consider a revised submission. Please carefully revise your manuscript in accordance with the reviewers' recommendations, particularly those of Reviewer #3, whose comments were more critical. Note that Reviewer #2' s comments were submitted as an attachment; please refer to the enclosed document for their detailed feedback. We emphasize that addressing these concerns is essential for further consideration of your work.

Please submit your revised manuscript within 60 days Jul 24 2025 11:59PM. If you will need more time than this to complete your revisions, please reply to this message or contact the journal office at plospathogens@plos.org. Please include the following items when submitting your revised manuscript:

We look forward to receiving your revised manuscript.

Kind regards,

Huiquan Liu

Guest Editor

PLOS Pathogens

Bart Thomma

Section Editor

Editor-in-Chief

PLOS Pathogens

Michael Malim

PLOS Pathogens

orcid.org/0000-0002-7699-2064

**Journal Requirements:**

https://journals.plos.org/plospathogens/s/submission-guidelines#loc-parts-of-a-submission

- ® on pages: 12, and 15

- TM on pages: 14, 15, and 22.

Potential Copyright Issues:

i) Figure 1C.  We note that the figure is created through BioRender. Please confirm that you hold a Premium account and provide a pdf copy of the CC BY 4.0 Licence as provided by BioRender. For instructions on how to generate a CC BY 4.0 license for your figure, please see the guidelines here: https://help.biorender.com/hc/en-gb/articles/21282341238045-Publishing-in-open-access-resources. 

If you are using the free assets from BioRender, we are unable to publish these images as they are licenced under a stricter licence than CC BY 4.0. In this case we ask you to remove the BioRender images and replace them with open source alternatives.

See these open source resources you may use to replace images / clip-art:

- https://bioart.niaid.nih.gov/ 

- https://bioicons.com/

- https://healthicons.org/ 

- https://scidraw.io/

- https://reactome.org/icon-lib

- https://www.phylopic.org/images 

- https://journals.plos.org/plosbiology/article?id=10.1371/journal.pbio.3002395

6) Thank you for stating "all data are included in the manuscript and supplementary material;proteomic data deposited at PRIDE." Please note that your Data Availability Statement is currently missing the DOI/accession number of each dataset OR a direct link to access each dataset. If your manuscript is accepted for publication, you will be asked to provide these details on a very short timeline. We therefore suggest that you provide this information now, though we will not hold up the peer review process if you are unable.

Note: Please ensure that the Data Availability statement provided in the online submission form matches the one included in the manuscript.

7) Please amend your detailed Financial Disclosure statement. This is published with the article. It must therefore be completed in full sentences and contain the exact wording you wish to be published.

3) If any authors received a salary from any of your funders, please state which authors and which funders.

Please include the grant number in the Financial Disclosure Field.

8) Please provide a completed 'Competing Interests' statement, including any COIs declared by your co-authors. If you have no competing interests to declare, please state "The authors have declared that no competing interests exist". Otherwise please declare all competing interests beginning with the statement "I have read the journal's policy and the authors of this manuscript have the following competing interests:"    . 

**Comments to the Authors:**

**Please note that one of the reviews is uploaded as an attachment.**

**Reviewers' Comments:**

Reviewer's Responses to Questions

**Part I - Summary**

Reviewer #1: In this paper, the authors show that the cellular localisation of the C. heterostrophus Hog1 stress activated protein kinase changes following ferulic acid (FA) stress, in that it forms discrete cytoplasmic foci. Interestingly, Hog1 is dephosphorylated in response to FA stress, but this seems dispensable for accumulation in such foci as is also seen in phosphatase mutants that block FA-induced Hog1 dephosphorylation. Evidence is also presented that such foci represent stress granules that are stimulated upon FA exposure, however the formation of such stress granules is Hog1 independent. So the strength of the study is the novel finding that FA stress drives Hog1 accumulation into stress granules. It is also noteworthy that proteomic approaches (by enriching the FA induced foci and analysing the proteome, and by determining the FA induced Hog1-interactome) and immunohistochemistry approaches characterised the contents of the FA-induced stress granules. Perhaps a weakness of the study is that the physiological role for this redistribution of Hog1 to cytoplasmic stress granules following FA stress remains unclear.

Reviewer #2: (No Response)

Reviewer #3: This study investigates the cytoplasmic sequestration mechanism of the stress-activated MAPK Hog1 in the maize fungal pathogen Cochliobolus heterostrophus under ferulic acid (FA) stress. Through fluorescence labeling, proteomic analysis, and co-localization experiments, the authors demonstrate that FA induces Hog1 to form cytoplasmic foci, which co-localize with RNA-binding proteins such as Puf2 within stress granules (SGs). The study reveals that Hog1 dephosphorylation and sequestration act in coordination, potentially preventing hyperactivation of signaling pathways and thereby enhancing fungal survival in the host environment. The FA-induced formation of Hog1 cytoplasmic foci and their colocalization with RNA-binding proteins, translation initiation factors, and mitochondrial components represents an interesting phenomenon. However, several experimental details require clarification, and mechanistic interpretations of some key conclusions remain incomplete. Additional validation experiments are necessary to enhance the study’s overall credibility.

**Part II – Major Issues: Key Experiments Required for Acceptance**

Reviewer #1: Considering the statement in the discussion; FA could be a cue to the pathogen that it needs to contend with host defenses including oxidative or osmotic stresses, both of which raise Hog1 phosphorylation. The observation (Fig S3B) that FA pre-treatment blocks the sorbitol induced nuclear accumulation of Hog1 is intriguing. What would be particularly informative is a western blot looking at Hog1-P over this sequential treatment. Does FA pre-treatment block the sorbitol-induced strong P of Hog1, or does Hog1-P still occur and the FA-induced cytoplasmic sequestering of Hog1 prevents nuclear accumulation. Irrespective of what scenario is correct it would be good to test whether FA pretreatment (and sequestering of Hog1) inhibits or promotes survival following stress treatment (osmotic stress/oxidative stress) that drives the phosphorylation and nuclear accumulation of Hog1.

The conclusion that FA dephosphorylation of Hog1 is not a pre-requisite for Hog1 sequestration into cytoplasmic granules is made based upon data presented in Fig. S4B. However, can the authors comment on whether dephosphorylation of Hog1 is important for resistance to FA stress?

Reviewer #2: (No Response)

Reviewer #3: 1. Regarding the abstract statement about Hog1 dephosphorylation being 'surprising,' this observation is in fact consistent with previous findings demonstrating ferulic acid (FA)-induced Hog1 dephosphorylation in C. heterostrophus (Environ Microbiol, 2016, 18:4188-4199). To better understand the functional significance of this response, the authors might consider examining: (1) whether FA triggers Hog1 accumulation in cytoplasmic foci, and (2) whether this dephosphorylation pattern persists during natural host infection. Such experiments could provide valuable insights into the role of this signaling pathway in host-pathogen interactions."

2. This manuscript contains multiple significant shortcomings. (1) The introduction section is currently too brief and does not adequately review the current scientific understanding of Hog1 signaling pathways. (2) Differential interference contrast (DIC) images are missing from all micrographs. (3) Scale bars are absent in Figure 4 and Supplementary Figures S1D-F and others. (4) Figure S2 requires reorganization for clarity, and the DAPI staining is particularly difficult to discern. To directly assess Hog1 mobility between nuclear and cytoplasmic compartments, fluorescence recovery after photobleaching (FRAP) experiments could be performed (5) Quantitative analyses (e.g., Fig. 1B and D) lack descriptions of the statistical methods used (e.g., t-test, ANOVA). The overall figure preparation appears rushed.

3. The SG-enriched fraction was isolated via differential centrifugation, but the specificity of this fraction is not validated. The authors should perform Western blot analysis to detect known SG markers to confirm successful SG enrichment.

4. The co-localization of Hog1 with Puf2 suggests association, but it is unclear whether Hog1 binds RNA directly or is recruited via protein-protein interactions. Additionally, since Hog1 foci still form in the Puf2 deletion mutant, the role of other SG components should be considered and discussed in the context of potential redundancy.

**Part III – Minor Issues: Editorial and Data Presentation Modifications**

Reviewer #1: It is not clear why so much of the data in the paper is in the Supplementary figures? For example, the immunohistochemistry images shown in Fig S4 beautifully show Hog1 cytoplasmic localisation and dephosphorylated state following FA treatment.

For the non-expert, it appears that the localisation of Hog1 demonstrates variation even between different filaments of C. heterostrophus, and in some places the description of the localisation pattern does not seem to match that presented. For example, an aim was to explore if Hog1 could initially be transported to the nucleus upon FA stress. However, in contrast to the conclusion that exposure to FA caused direct Hog1:GFP localization to bright cytoplasmic foci, starting within 10 minutes, in the 10 min FA images there is co-localisation of Hog1-GFP with the DAPI stain (Supp Fig 2)? In a further example in Fig S4B, the FA-mediated Hog1 localisation does not look like that in wild-type cells in any of the three mutants tested (Puf2, PtcB, Cdc14). Some comments in the text to support why the authors are confident in their conclusions would be beneficial.

The legends for Supp Fig.1 parts E and F should be aligned to the main legend.

What is the difference between the two FA data points on Supp Fig 1A?

Figure 1F. The protein samples from the fractionation experiment following control or FA treatment need to be processed on the same blot (currently shown as 2 panels) to allow direct comparison.

Whilst the experiments to explore potential Hog1-GFP co-localisation with mitochondria was supported by Co-IP of Hog1 and mitochondrial proteins, it is not clear why the authors explored whether Hog1-GFP foci colocalised with peroxisomes?

The Discussion is slightly disjointed – and a model figure summarizing the key findings and unanswered questions would be good.

Reviewer #2: (No Response)

Reviewer #3: 5. Although Fig. 4A shows Hog1 co-localization with mRNA foci, quantitative analysis is missing. At least three independent biological replicates should be quantified to enhance the reliability of this conclusion.

6. The study does not explore whether Hog1 sequestration affects downstream signaling. e.g., nuclear translocation of Hog1-dependent transcription factors. Expression analyses of Hog1 target genes, particularly oxidative stress-responsive genes, under FA treatment would help clarify functional consequences of Hog1 localization.

7. Fungal strain names should be italicized. Gene and protein nomenclature is inconsistent (e.g., "Hog1" vs. "ChHog1"). The entire manuscript should consistently use "ChHog1" to reflect species-specific gene naming conventions.

PLOS authors have the option to publish the peer review history of their article (what does this mean? ). If published, this will include your full peer review and any attached files.

**Do you want your identity to be public for this peer review?** For information about this choice, including consent withdrawal, please see our Privacy Policy .

Reviewer #1: No

Reviewer #2: No

Reviewer #3: No

**Figure resubmission:**

**Reproducibility:**



---

## [Decision Letter · Decision Letter 1]

1 Sep 2025

PPATHOGENS-D-25-00746R1

Cytoplasmic sequestering of a fungal stress-activated MAPK in response to a host plant phenolic acid

PLOS Pathogens

Dear Dr. Horwitz,

Thank you for submitting your manuscript to PLOS Pathogens. After careful consideration, we believe that your study shows merit but requires minor revisions to fully meet our publication criteria. We invite you to submit a revised version of the manuscript that addresses the minor points raised by the reviewers during the review process.

Please submit your revised manuscript within 30 days Oct 31 2025 11:59PM. If you will need more time than this to complete your revisions, please reply to this message or contact the journal office at plospathogens@plos.org. Please include the following items when submitting your revised manuscript:

We look forward to receiving your revised manuscript.

Kind regards,

Huiquan Liu, Ph.D.

Guest Editor

PLOS Pathogens

Bart Thomma

Section Editor

PLOS Pathogens

Sumita Bhaduri-McIntosh

Editor-in-Chief

PLOS Pathogens

orcid.org/0000-0003-2946-9497

Michael Malim

Editor-in-Chief

PLOS Pathogens

orcid.org/0000-0002-7699-2064

**Journal Requirements:**

1) We note that Legend for Suppl Video 1, and and Legend for Suppl Video 2 files are duplicated on your submission. Please remove them from the online submission form as the legends of the supporting information files should be included only in the manuscript.

-TM on page 15.

3) Thank you for stating "The mass spectrometry proteomics data have been deposited to the ProteomeXchange Consortium via the PRIDE partner repository with the dataset identifier PXD044457." Please note that, though access restrictions are acceptable now, your entire minimal dataset will need to be made freely accessible if your manuscript is accepted for publication. This policy applies to all data except where public deposition would breach compliance with the protocol approved by your research ethics board.

**Reviewers' Comments:**

Reviewer's Responses to Questions

**Part I - Summary**

Reviewer #1: In response to the previously raised concern that the physiological role of FA-induced cytoplasmic foci was lacking, the authors present new data showing that FA treatment prevents activation of the Hog1 SAPK in response to osmotic stress, thus supporting a hypothesis that FA production by the plant may block Hog1 activation during infection and thus could represent a mechanism to inhibit virulence.

Reviewer #2: This study addresses a novel aspect of host-pathogen interactions by uncovering a unique regulatory mechanism of the stress-activated MAPK Hog1 in Cochliobolus heterostrophus (causal agent of maize southern leaf blight) in response to ferulic acid (FA), a key plant phenolic defense compound. Its strengths are notable:

1.Fig. 4B reports “fraction colocalized” but lacks details on statistical methods (e.g., Manders’ overlap coefficient, which is standard for fluorescence colocalization). Adding these values (with SEM and p-values) would quantify the extent of Hog1-mRNA/Puf2 colocalization.

2.Fig. 1E/F conflates “SGEF” and “supernatant” fractions in the legend; clarify fraction names and ensure consistency with the centrifugation scheme (Fig. 1C). Scale bars in Fig. S1D–F and Fig. 4A are occasionally missing or inconsistent—standardize to 1–5 μm for clarity.

3.Experiments to measure Hog1 turnover or retention in SGs (e.g., fluorescence recovery after photobleaching [FRAP] in ptcB mutants vs. wild-type) could test if dephosphorylation affects Hog1’s dynamics in SGs.

4.Some references (e.g., ref 17: Hsiao et al.) lack complete journal information; ensure adherence to PLOS Pathogens’ reference style (e.g., full journal title, volume, pages).

Reviewer #3: The authors have revised and responded well to my concerns. The revised manuscript shows more improved and logical data.

**Part II – Major Issues: Key Experiments Required for Acceptance**

Reviewer #1: New data is presented in response to previous major issues.

Reviewer #2: (No Response)

Reviewer #3: (No Response)

**Part III – Minor Issues: Editorial and Data Presentation Modifications**

Reviewer #1: There are still parts of the discussion that are confusing at least to this reviewer.

For example the statement 'FA could be a cue to the pathogen that it needs to contend with host

defenses including any stresses that raise ChHog1 phosphorylation' its not clear how FA mediated dephosphorylation of Hog1 and cytoplasmic sequestration alert the pathogen to stresses that require Hog1 activation?

Figure S4 is missing details as to which panels represent Wt cells, and those which represent the different mutants.

Reviewer #2: (No Response)

Reviewer #3: (No Response)

PLOS authors have the option to publish the peer review history of their article (what does this mean? ). If published, this will include your full peer review and any attached files.

**Do you want your identity to be public for this peer review?** For information about this choice, including consent withdrawal, please see our Privacy Policy .

Reviewer #1: No

Reviewer #2: No

Reviewer #3: No

**Figure resubmission:**
---

## [Editor Report · Decision Letter 2]

13 Oct 2025

Dear Prof. Horwitz,

We are pleased to inform you that your manuscript 'Cytoplasmic sequestering of a fungal stress-activated MAPK in response to a host plant phenolic acid' has been provisionally accepted for publication in PLOS Pathogens.

Best regards,

Bart Thomma

Section Editor

PLOS Pathogens

Sumita Bhaduri-McIntosh

Editor-in-Chief

PLOS Pathogens

orcid.org/0000-0003-2946-9497

Michael Malim

Editor-in-Chief

PLOS Pathogens

orcid.org/0000-0002-7699-2064
---

## [Editor Report · Acceptance letter]

Dear Prof. Horwitz,

We are delighted to inform you that your manuscript, "Cytoplasmic sequestering of a fungal stress-activated MAPK in response to a host plant phenolic acid," has been formally accepted for publication in PLOS Pathogens.

Best regards,

Sumita Bhaduri-McIntosh

Editor-in-Chief

PLOS Pathogens

orcid.org/0000-0003-2946-9497

Michael Malim

Editor-in-Chief

PLOS Pathogens

orcid.org/0000-0002-7699-2064